# Using Radar Observations to Evaluate 3D Radar Echo Structure Simulated by the Global Model E3SM Version 1

Jingyu Wang[1], Jiwen Fan[1,*], Robert A. Houze Jr [2], Stella R. Brodzik[2], Kai Zhang[1], Guang J. Zhang[3], and Po-Lun Ma[1]

[1]Pacific Northwest National Laboratory, Richland, WA 99354, USA
[2]University of Washington, Seattle, WA 98195, USA
[3]Scripps Institution of Oceanography, La Jolla, CA 92093, USA

*Correspondence to*: Jiwen Fan (jiwen.fan@pnnl.gov)

**Abstract.** The Energy Exascale Earth System Model (E3SM) developed by the Department of Energy has a goal of addressing challenges in understanding the global water cycle. Success depends on correct simulation of cloud and precipitation elements. However, lack of appropriate evaluation metrics has hindered the accurate representation of these elements in general circulation models. We derive metrics from the three-dimensional data of the ground-based Next generation radar (NEXRAD) network over the U.S. to evaluate both horizontal and vertical structures of precipitation elements. We coarsened the resolution of the radar observations to be consistent with the model resolution and improved the coupling of the Cloud Feedback Model Intercomparison Project Observation Simulator Package (COSP) and E3SM Atmospheric Model Version 1 (EAMv1) to obtain the best possible model output for comparison with the observations. Three warm seasons (2014-2016) of EAMv1 simulations of 3D radar reflectivity features at an hourly scale are evaluated. A general agreement in domain-mean radar reflectivity intensity is found between EAMv1 and NEXRAD below 4 km altitude; however, the model underestimates reflectivity over the central United States, which suggests that the model does not capture the mesoscale convective systems that produce much of precipitation in that region. The shape of the model estimated histogram of subgrid scale reflectivity is improved by correcting the microphysical assumptions in COSP. Different from previous studies that evaluate modelled cloud top height, we find the model severely underestimates radar reflectivity at upper levels—the simulated echo top height is about 5 km lower than in observations—and this result is not changed by tuning any single physics parameter. For more accurate model evaluation, a higher-order consistency between the COSP and the host model is warranted in future studies.

## 1 Introduction

Clouds and precipitation play a major role in Earth's budgets of energy, water, and momentum. However, the correct simulation of 3D structures of clouds and precipitation has been challenging in general circulation models (GCMs) (Trenberth et al., 2007; Randall et al., 2007; Eden and Widmann, 2012), partially because model grid spacings generally do not adequately resolve the cloud-structure details important to these budgets. In addition, the lack of appropriate evaluation metrics also

hinders the evaluation of GCMs. Over the contiguous U.S. (CONUS), the detailed 3D radar reflectivity field (indicating the 3D distribution of precipitation particles) is observed by the ground-based Next Generation Radar (NEXRAD) network of S-band weather radars (3 GHz; Zhang et al., 2011 and 2015). In this study, we use the mosaic of NEXRAD observations called Gridded Radar Data (GridRad) developed by Homeyer and Bowman (2017), which have a horizontal resolution of 0.02°

(regridded to 4 km in this study), vertical resolution of 1 km (24 levels), and an update cycle of 1 hour. In order to compare these data appropriately with output of the global model used here, we further coarsen the horizontal resolution, as described in Section 2.

The Energy Exascale Earth System Model (E3SM) is an ongoing effort of the Department of Energy (DOE) to advance the next-generation of climate modeling (Bader et al., 2014). Version 1 of E3SM Atmosphere Model (EAMv1) is a descendent of

the National Center for Atmospheric Research (NCAR) Community Atmosphere Model version 5.3 (CAM5.3; Neale et al., 2012). However, it has evolved substantially in coding, performance, resolution, physical processes, testing and development procedures (Rasch et al., 2019). Previous model evaluation has focused on the long-term climatological properties of certain cloud fields, surface precipitation, and water conservation on the global scale (e.g., Qian et al., 2018; Xie et al., 2018; Zhang et al., 2018; Lin et al., 2019). Evaluations of the vertical structures of cloud and precipitation elements have used vertically

pointing radar observations obtained during field campaigns (Zhang et al., 2018; Zhang et al., 2019). However, these tests lacked evaluation of fully 3D cloud and precipitation structure over large regions of the globe and over long time periods.

For this study, we have built data processing techniques to evaluate EAMv1 simulation of the 3D radar reflectivity field at its default setting of 1° grid spacing and 72 vertical layers at an hourly time scale. Our goal is to provide a comprehensive evaluation of both horizontal pattern and vertical structure of cloud and precipitation. We use radar observations obtained from

the NEXRAD over the CONUS for the three years 2014-2016. In order to directly compare the model results with NEXRAD, we have implemented and improved the Cloud Feedback Model Intercomparison Project (CFMIP) Observation Simulator Package (COSP) (Bodas-Salcedo, et al., 2011) into EAMv1. We restrict the evaluation to the warm season (i.e., April to September). Over the CONUS, warm season precipitation is dominated by convective processes, which are very different from the more widespread frontal cloud systems of cold-season precipitation. As discussed by Iguchi et al. (2018), precipitating ice

particles have large variation in habits and scattering properties, and the effect of non-Rayleigh scattering and multiple scattering by large precipitating ice particles could introduce large uncertainty into simulating the radar reflectivity field. To reduce uncertainty due to these factors, we examine only the warm season of the three years from 2014 to 2016.

This paper is organized as follows: Section 2 describes the model, the GridRad dataset, the COSP simulator, and the step-by-step methodology of data processing to account for differences between the modelled and observed datasets, specifically (1)

horizontal and vertical resolutions of EAMv1 (1°, 72 vertical levels) and NEXRAD (4 km horizontally, 1 km vertically) and (2) minimum detectable limits between the model and NEXRAD. Section 3 presents the model evaluation results and tests of the sensitivity to physics parameters. Section 4 provides synthesis and conclusions.

## 2 Methodology

### 2.1 EAMv1 Description and Configuration

EAMv1's dynamics core and physics parameterizations are described in detail by Rasch et al. (2019). The continuous Galerkin spectral finite element method solves the primitive equations on a cubed-sphere grid (Dennis et al., 2012; Taylor & Fournier, 2010). Tracer transport on the cubed sphere is handled using a variant of the semi-Lagrangian vertical coordinate system of Lin (2004). The method locally conserves air mass, trace constituent mass, and moist total energy (Taylor, 2011). Turbulence, shallow cumulus clouds, and cloud macrophysics are parameterized with the Cloud Layers Unified By Binormals (CLUBB)

parameterization (Golaz et al., 2002; Larson, 2017). Deep convection is based upon the formulation originally described in Zhang and McFarlane (1995, hereafter ZM), with modifications by Neale et al. (2008) and Richter and Rasch (2008). Stratiform clouds are represented with the "Morrison and Gettelman version 2" (MG2) two-moment bulk microphysics parameterization (Gettelman and Morrison, 2015). Aerosol microphysics and interactions with stratiform clouds are treated with an updated and improved version of the four-mode version of the Modal Aerosol Module (MAM4; Liu et al., 2016).

Regarding the stratiform-convection partition, the MG2 stratiform cloud microphysics and CLUBB higher-order turbulence parameterization explicitly provide values for condensate mass and number, as well as an estimate of stratiform cloud fraction, whereas the convective cloud fraction is not parameterized in mass flux-based ZM scheme (assumed to be <<1 for typical GCM resolutions such as at 1-degree grid spacing or coarser), and is diagnosed from cloud mass flux for cloud radiation calculation, which is treated as a tunable parameter.

The EAMv1 used in this study has 30 spectral elements (ne30), which corresponds to approximately 1° horizontal grid spacing, and the total number of grid columns is 48,602. Vertically, there are 72 layers using a traditional hybridized sigma pressure coordinate. The simulation is run for the time period from 1 January 2014 to 1 October 2016. We use a dynamic timestep of 5 min and a cloud microphysics timestep of 30 min. The large-scale circulation in the simulation is constrained using the nudging technique (Zhang et al., 2014; Ma et al., 2015; Lin et al., 2016), so that the model simulations can be constrained by realistic

large-scale forcing. Specifically, horizontal winds (U, V components) are nudged towards the Modern-Era Retrospective analysis for Research and Applications, Version 2 (MERRA2) reanalysis data (Gelaro, et al., 2017) with a relaxation time scale of 6 hours. Nudging is applied to all grid boxes at each time step, with the nudging tendency calculated using the model state and the linearly-interpolated MERRA2 data (Sun et al., 2019).

    To facilitate the comparison with observations, model outputs are regridded to the geographic coordinate system with a

horizontal grid spacing of 100 km, and the vertical coordinate is converted to the above mean surface level height in meters. By default, all the regridding processes in this study are based on the Earth System Modeling Framework (ESMF) Python Regridding Interface (https://www.earthsystemcog.org/projects/esmpy/) using bilinear interpolation.

**2.2 COSP Radar Simulator**

The retrieved spaceborne satellite and ground-based radar products such as cloud water content, and effective particle size
(e.g., Randel et al., 1996; Wang et al., 2015; Tian et al., 2016; Um et al., 2018) are often treated as the ground-truth for model evaluation (e.g., Fan et al., 2017; Han et al., 2019). However, the retrieved products often have large uncertainty (Stephens and Kummerow, 2007). To allow the comparison of model results with direct measurements from 3D scanning radars (ground-based or satellite-borne), the CFMIP Observation Simulator Package (COSP) was developed for use in GCMs (Bodas-Salcedo et al., 2011). Instead of using retrieved products to evaluate the model simulation, COSP converts model output into pseudo-
observations using forward calculations (Bodas-Salcedo et al., 2011; Swales et al., 2018; Zhang et al., 2010).

The COSP consists of three steps, as detailed in Zhang et al. (2010). The first step is to generate a subgrid-scale distribution of cloud and precipitation, which is done by using the Subgrid Cloud Overlap Profile Sampler (SCOPS; Klein and Jakob, 1999; Webb et al., 2001) and SCOPS for precipitation (SCOPS_PREC), respectively. Each GCM grid box is divided into 50 subcolumns in this study. Detailed description of SCOPS and SCOPS_PREC can be found in Zhang et al. (2010). Then, the
radar signals are calculated by the QuickBeam code (Haynes and Stephens, 2007) using the column distribution of cloud and precipitation. Thus, COSP calculates the reflectivity for the combined cloud properties using its own subgrid assumption, and it does not distinguish convective and stratiform cloud contributions to reflectivity. Finally, the grid box mean radar reflectivity is calculated through the method of linear averaging (i.e., the reflectivity values [in dBZ] are converted to the Z values [mm$^6$ m$^{-3}$] to calculate the mean Z, then mean Z is converted back to the dBZ). In addition to averaging, all the processing of radar
reflectivity data from model and NEXRAD in this study utilizes the linearized Z values, including horizontal averaging, vertical interpolation, calculation and comparison of mean values, etc.

The COSP version 1.4 used in this study has no scientific difference from version 2.0 (Song et al., 2018, Swales et al., 2018). Following the general usage of COSP, we modified the microphysics assumptions used for the radar reflectivity calculation regarding hydrometeor density, size distribution, etc., making those assumptions consistent with those used in the MG2 cloud
microphysics scheme that is used in E3SM. The detailed documentation of those changes is in Table 1. Note that, although we tried to make the COSP use the same hydrometeor size distribution functions as MG2, the three parameters (slope, intercept, and shape parameters) are still separately defined in COSP. We use horizontally homogeneous cloud condensate distribution within the model grid element, and maximum-random overlapping scheme for cloud occurrence (Marchand et al., 2009; Hillman et al., 2018).

**2.3 NEXRAD Observations**

The NEXRAD network consists of 159 S-band (3 GHz) Doppler radars, which form a dense observational network nearly covering the CONUS. We use the GridRad mosaic product of Homeyer and Bowman (2017), which combines all NEXRAD radar data covering the region 155°W – 69°W, 25°N – 49°N. To compare the GridRad data to the E3SM model fields, the radar frequency in the COSP was set to 13.6 GHz, consistent with the Global Precipitation Measurement (GPM) Ku-band

radar, since we originally aimed at evaluating the E3SM simulation with GPM data. However, due to the high detectable threshold of 13 dBZ, low sampling frequency (4-7 overpasses over CONUS per day), and the narrow swath width (245 km) for each overpass, GPM data within the three-year period (2014-2016) have a significant under-sampling issue. That is, the GPM sample sizes over 1° model grid boxes are generally too small to robustly represent the grid element mean value. Therefore, we decided not to use GPM data in this study. As GPM operates over the whole earth and is anticipated to run for

a long-time period, it will likely be a very useful dataset to evaluate the coarse-resolution global model in the future.

The GPM radar frequency is higher than the NEXRAD (13.6 GHz vs. 3 GHz). Previous studies have shown conversions from Ku (13.6 GHz) to S band (3 GHz) are necessary when using GPM Ku band radar to calibrate the ground-based radars (Warren et al., 2018). Based on our previous study that quantitatively evaluated the coincident observations from NEXRAD and GPM over the CONUS, we found the 3D radar reflectivity fields obtained from the two independent platforms are highly consistent

with each other after proper smoothing of GPM data in the vertical (Wang et al., 2019b). We performed a series of offline tests of COSP simulation using the frequency of 3 GHz (NEXRAD), 13.6 GHz (GPM Ku band), and 94 GHz (the cloud profiling radar onboard of the CloudSat satellite). Their corresponding reflectivities are compared in Fig. 1. As shown, the reflectivity values with 3 GHz are very similar to those with 13.6 GHz, indicating the Rayleigh scattering is satisfied for both frequencies in this application. To examine if the COSP can correctly handle the Mie scattering calculation, the frequency of 94 GHz used

by the CloudSat is also tested, whose products have been widely used for the evaluation of coarse-resolution models (Zhang et al., 2010). As shown in Fig. 1, the reflectivities simulated with 94 GHz significantly deviate from those simulated with 3 GHz and 13.6 GHz when reflectivities > 10 dBZ, which reveals that the COSP simulator is capable of handling both Rayleigh and Mie scattering calculations. However, there is no difference using Ku band or S band in the COSP simulator in this study, because the simulated condensates are not large enough to lead to non-Rayleigh scattering, which is typically observed at $Z >$

dBZ for the Ku-band (Matrosov, 1992).

An attenuation correction has been applied in case of existence of any large particles although they are extremely unlikely to occur in this application. Since the COSP mimics the satellite view from space to the ground, the layer below 1-km altitude is most vulnerable to the possible attenuation caused by large precipitation particles, which has been excluded from the comparison. In this study, biases caused by the temporal mismatch are minimal at the horizontal resolution of 1° (~100 km),

we nevertheless perform Gaussian smoothing of GridRad data to match the model time step (30 min) in the comparison.

**2.4 Mapping the Radar Observations to the Model Grid**

As shown in previous studies (e.g., Wang et al., 2015, 2016, 2018; Feng et al., 2012, 2019), the minimum reflectivity of the 3D mosaic NEXRAD dataset is 0 dBZ (Fig. 2a). However, the model grid-mean reflectivity can be as low as -100 dBZ.

Because our focus is on significantly precipitating clouds, the minimum threshold of reflectivity at 1° grid scale is set to be 8 dBZ (corresponding to rain rate $\geq 0.1$ mm hr$^{-1}$). We also tested with a threshold of 0 dBZ and report later on how it only has minor effects on our conclusions. For our main results, after coarsening the 4-km GridRad data to a model grid element, only

the grid elements with a mean value larger than 8 dBZ are taken into account in both observations (Fig. 2b) and in the simulation (Fig. 2c). In the vertical direction, the EAMv1-simulated radar reflectivity field (72 vertical levels, hybrid coordinate) is interpolated to the levels of GridRad (vertical resolution of 1 km). The simulation data are saved hourly, consistent with the hourly GridRad data.

## 3 Results

After the horizontal averaging, vertical interpolation, and truncation at the identified minimum threshold of 8 dBZ, the 3D radar reflectivity fields obtained from GridRad and the model simulation become comparable. The EAMv1 simulated reflectivity is evaluated from the perspectives of subgrid distribution, horizontal pattern, and vertical distribution.

### 3.1 Comparison on Subgrid Distribution of Reflectivity

The horizontal resolution difference between GCMs (~100 km) and NEXRAD observations (4 km) presents a challenge for testing the model simulated radar reflectivity. To mimic the observations, COSP divides the grid-mean cloud and precipitation properties into subcolumns (Pincus et al., 2006) that statistically downscale the data in a way that should be consistent with observations. The way this is done in COSP is discussed by Zhang et al. (2010) and Hillman et al. (2018). In this section we examine whether the subgrid reflectivity distribution generated by COSP is consistent with the observed subgrid reflectivity distribution shown by the NEXRAD observations.

In EAMv1, 50 subcolumns are used for calculating the mean radar reflectivity for a model grid box. There are 625 pixels inside each 1° grid for NEXRAD data to provide a probability density function (PDF) of observed reflectivity within the box. After averaging the NEXRAD pixels at subgrid scale to 50 samples to match the COSP's subcolumns, Fig. 3 compares the simulated subgrid reflectivity PDF to the NEXRAD PDF based on all the GridRad samples combined for the 3-year period at each individual level, where the interval of reflectivity bins is 1 dBZ. The results for the default microphysics assumptions in COSP, which are for a single-moment scheme, produce a bi-modal distribution at and below 8-km altitudes (blue histograms in the left-hand column of Fig. 3). The bimodality is significantly different from the observed PDF, which forms a smooth gamma distribution. Song et al. (2018) also found bimodal distributions when the COSP was implemented in the CAM with the original microphysics assumptions, which are clearly unlike real observed radar reflectivity distributions.

Our modification of the microphysical assumptions in COSP (right-hand column of Fig. 3) greatly reduces the bimodality. In addition, the modified microphysical assumptions produce higher values of reflectivity, in better agreement with observations, and the grid-mean radar reflectivities increase by ~4 dBZ (Fig. 4) mainly for values less than 25 dBZ. The improvement in the subgrid distribution and grid-mean reflectivity brought by the change of microphysics assumptions indicates the necessity of microphysical consistency between the COSP and the host model. It should be noted that the simulated radar reflectivity and its subgrid distribution are sensitive to the overlap assumption and the distribution function of condensates that are set in COSP

(Hillman et al., 2018). Our results are from the default setup of these aspects of COSP. It is not the purpose of this study to test those assumptions.

Although the simulated subgrid reflectivity distribution is improved by setting the microphysics assumptions used in COSP consistent with the MG2, the model is still significantly biased. In addition to the intrinsic model-observation differences in the number concentrations and mixing ratios of hydrometeors, there are other possible error sources related to the reflectivity calculation as mentioned in Section 2.2. For example, (1) the mixing ratios of hydrometeor types from different types of clouds are not directly passed from the host model to COSP, instead they are lumped together and equally divided among all the

precipitating subcolumns, (2) the spectral parameters for defining a Gamma distribution are not consistent from MG2, and (3) the assumptions of subgrid distribution and hydrometeor vertical overlap are simple and not consistent with other parts of the host model. In addition, the subgrid distribution results from COSP are calculated based on the assumption about the distribution of cloud and precipitation among the 50 subcolumns, which is independent of what E3SM uses. Therefore, a higher-order consistency between the COSP and the host model is warranted in future studies.

In this following analysis, we focus on the evaluation of the simulated 3D radar reflectivity field at the model's native grid, which is 1°, since the subgrid information from COSP does not directly reflect how E3SM does it. Also, the convective cloud fraction is not parameterized in mass flux-based ZM scheme and is diagnosed from cloud mass flux for cloud radiation calculation, which is treated as a tunable parameter, whose evaluation is not very meaningful unless it becomes an independent variable, for instance, for grey-zone resolutions.

**3.2 Comparison of Horizontal Patterns**

Now we compare the temporal mean reflectivity through the entire study period between the NEXRAD observation (Figs. 5a, d, g and j) and EAMv1 simulation (Figs. 5b, e, h, and k) with the consistent microphysical assumptions between COSP and the host model at the vertical levels of 2, 4, 8, and 11 km. The mean, standard deviation, 95th percentile values, and valid sample numbers between the model and NEXRAD are compared in Table 2. At 2-km altitude, the EAMv1 estimates higher

reflectivity than the NEXRAD observations (Figs. 5a-b) except over the central United States. The overall mean value is 28.7 dBZ for EAMv1 and 25.1 dBZ for NEXRAD. The negative bias for the model is in the region between the Rocky Mountains and Mississippi basin (Fig. 5c), where precipitation is heavily contributed by Mesoscale Convective Systems (MCSs). Those MCSs propagate eastward from their initiation over or just east of the Rocky Mountains, go through upscale growth, and finally dissipate in the eastern part of the Mississippi Basin (Yang et al. 2017; Feng et al., 2018, 2019). The standard deviations

of the two individual datasets are quite similar, and EAMv1 generates a higher 95th percentile value than the observation, indicating the model overestimates the extreme high values at lower troposphere. In addition, those simulated extreme values are evenly distributed across the entire domain, which fail to mimic the spatial footprint of MCSs as depicted by the NEXRAD data.

At 4-km altitude (Figs. 5d-e), the model's underestimation over central U.S. becomes larger compared to the 2-km altitude

and the overestimation at the foothills of Rocky Mountains also becomes larger. The model also overestimates reflectivity in

the east region of the domain. These results indicate that the E3SM simulation fails to capture the observed spatial variability. The domain mean value between the model and observations is the same (24.0 dBZ) as a consequence of the offset between the negative and positive biases in different areas. The standard deviation and 95th percentile values are comparable with the observations as well. At 8 km, underestimation of the reflectivity by the model occurs over almost the entire domain (Fig. 5i),

with a domain mean of 15.0 dBZ, much lower than 19.2 dBZ in the NEXRAD data. Meanwhile, the modelled standard deviation and the extreme values are smaller, indicating the model has a difficulty capturing the observed variability.

At 11-km altitude, the EAMv1 severely underestimates the reflectivity values compared to NEXRAD (Figs. 5j-k), with a mean value of 9.8 dBZ for EAMv1 while 16.6 dBZ for NEXRAD. The negative bias is generally more than 7.5 dBZ in the central United States (Fig. 5l), and the model severely underestimates the standard deviation and extreme reflectivity. Moreover,

EAMv1's sample size is 50 time lower than that of the NEXRAD, indicating the lower occurrence of reflectivity values ≥ 8 dBZ.

Clearly, above 4 km, the model's negative biases increase with height as shown from Figs. 5f, i, and l, manifested in the central United States. There is no valid reflectivity value simulated by EAMv1 above 12-km altitude, where NEXRAD still shows reflectivity values up to 15.7 dBZ, indicating that the simulated deep convection in the warm season is not deep enough, a

problem that is further examined in the following section.

In addition to the mean values, the histograms of observed and simulated radar reflectivities are compared for different altitudes, where the interval of reflectivity bins is 2 dBZ (Fig. 6). By comparing the occurrence of Z ≥ 8 dBZ between model and observations, the model apparently has narrower distribution than the observations, and the model-observation deviation in maximum values increases with height. At 8 km and below, the model generally overestimates the sample sizes of smaller

reflectivity values but lacks extreme high reflectivity values. However, at 11-km altitude, the model greatly underestimates the sample sizes of the entire reflectivity spectrum compared to the observation, causing the severe underestimation in the mean value.

## 3.3 Comparison of Vertical Distribution of Radar Reflectivity

To quantitatively examine the simulated vertical distribution of radar reflectivity, contoured frequency by altitude diagrams

(CFADs, Yuter and Houze 1995) are generated from NEXRAD and EAMv1 and compared in Fig. 7. The CFADs represent the frequency of occurrence of reflectivity in a coordinate system having reflectivity bins (interval of 1 dBZ) on the x-axis and altitude bins (interval of 1 km) on the y-axis. The frequency within each bin box is calculated as the number of valid samples it contains divided by the total sample number of all reflectivity bins at all levels, meaning that the integrated value of all frequencies in each plot is 100%.

Fig. 7 shows the CFADs for both NEXRAD observations (Figs. 7a, d, g, j, m, and p) and the EAMv1 simulation (Figs. 7b, e, h, k, n, and q) for each month from April to September combined over 2014-2016. The most distinct difference between the model and observations is the simulated echo top height. The echo top height in the simulation generally is at 11 km, at least 5 km lower than the 16 km top seen in the observations. At levels below 4 km, the NEXRAD data show a high frequency zone

(> 3.2%) concentrated between 8-25 dBZ, whereas the simulated high frequency zone is at 13-28 dBZ. For reflectivity > 35

dBZ, the simulation has higher probability of occurrence than the NEXRAD observations.

Regarding the overall shape of CFADs, the model follows the well-known pattern where the reflectivity value range of high frequency zone (> 3.2%) increases from cloud top to the freezing level, and then slowly decreases or remains constant below the freezing level. The cores of maximum frequency (> 5%) are located in the centres of the high frequency zones. However, these characteristics are not presented in the observations, whose high frequency zones are greatly skewed to the lower

reflectivity values. The characteristics of NEXRAD's CFADs could be due to averaging from fine resolution (4 km) to coarse resolution (1°), as well as averaging of convective and stratiform components because the two components produce significantly different reflectivity profiles and magnitudes.

The box-whisker plots (Figs. 7c, f, i, l, o, and r) represent the same results in a different way, where the normalization is conducted at each level rather than against the entire dataset at all levels. Below 4 km, the percentile values are consistent

between the model and observations except for the 1-km altitude where the model overestimates the reflectivity. The simulated 25-75th percentiles are located at the reflectivity values of 15-27 dBZ at 1-km altitude, which is higher than the NEXRAD observation (12 - 28 dBZ). As noted in the discussion of Fig. 5, the consistency at low-levels (e.g., 2 km) between the model and observations is mainly due to the offset of negative and positive biases at different regions of the domain. Moreover, EAMv1 underestimates the frequency of echoes ≤ 15 dBZ and overestimate it for echoes between 15 and 30 dBZ, which

causes the higher median values in the model. From 4 km upward, the model-observation differences become much larger than at low levels, consistent with the result shown in Fig. 5. The underestimation of 95th percentile value increases from 10 dBZ at 7 km to more than 20 dBZ at 11 km. Above 11 km, the model fails to generate average reflectivity above 8 dBZ, and the typical reflectivity value is between 0 and 2 dBZ at 12 km.

From Fig. 7 it is clear that the model severely underestimates the echo top height by at least 5 km. To look at how this result

is sensitive to the threshold reflectivity, we reprocessed the results with the 0 dBZ threshold. By lowering the threshold to 0 dBZ, an increment of ~1 km in the vertical extension of the CFADs is found in the model, but the echo top height of the observations is not changed much. As a result, the choice of threshold does not change the conclusion of severe model underestimation in echo top height.

The CFADs of NEXRAD observations vary from month to month. For example, the echo top height is at 15 km in April,

which increases to 16 km in May, then reaches 17 km in June and July, and finally decreases to 15 km in September. Similarly, the 0.6%-0.8% contour level in the observations stops at 9-km altitude in April, but extends to 10 km in May and reaches 11 km in June. It increases to the highest at 11.5 km in July and August, then decreases to 11 km in September. This seasonality follows the seasonal variation of intensity of convection (Wang et al., 2019a), which is not captured in the EAMv1 simulation (Figs. 7b, e, h, k, n, and q).

The severe underestimation of the echo top height by EAMv1 has been reported for simulation of tropical convection with the Community Atmosphere Model version 5 (CAM5) in a recent study (Wang and Zhang, 2019). Although EAMv1 is different from CAM5 in many aspects such as vertical resolution and dynamical core, they share the same Zhang-McFarlane (ZM)

cumulus parameterization (Zhang and McFarlane, 1995) for representing deep convection. Wang and Zhang (2019) found the cloud top height of tropical convection is underestimated by more than 2 km, which can be alleviated by the adjustment of the

ZM scheme. We have performed a series of sensitivity tests by changing physical parameters in ZM and cloud microphysics schemes to explore the possibility of model improvement in echo top height. These tests are detailed in Section 3.4.

As evaluated in Zheng et al. (2019), E3SM v1 failed to simulate the diurnal variation of precipitation over the central United States, where the observed nocturnal peak is greatly underestimated. Xie et al. (2019) improved the diurnal cycle of convection in E3SM v1 recently by modifying convective trigger function in the ZM scheme. It will be interesting to see if the 3D radar

reflectivity fields can be better simulated using the updated ZM scheme.

## 3.4 Sensitivity of Simulated Echo Top Height to Tunable Parameters of the Global Model

Differently from the model evaluation of cloud top height and high cloud fraction (e.g., Xie et al., 2018), where EAMv1 has shown good agreements with satellite observations over the CONUS, evaluation of radar echo top height indicates whether the processes internal to the cloud are producing precipitation correctly. To examine if any model parameters in the ZM

cumulus parameterization scheme and/or MG2 microphysics parameterization scheme can significantly influence the echo top height, we conducted a series of sensitivity tests for the tunable parameters as listed in Table 3. In each test a single parameter is changed, and all other parameters retain their default values.

Wang and Zhang (2018) suggested that the restriction of neutral buoyancy level (NBL) from the dilute CAPE calculation (Neale et al. 2008) can limit the depth of deep convection in ZM. When the convective plume reaches the NBL, all mass flux

is detrained even if the updraft is still positively buoyant from the cloud model calculation (Zhang, 2009). To allow deep convection to grow deeper, we performed a sensitivity test following Wang and Zhang (2018), where the NBL determined in the dilute CAPE calculation is removed, and the upper limit of the integrals of mass flux, moist static energy, and other cloud properties is set to be very high (70 hPa in this study). After the modification, the convective cloud top height increases as shown in Wang and Zhang (2018), however there is no change in the radar echo top height, i.e., the maximum altitude at which

precipitation-sized particles occur. A possible reason for the limited effect on echo top height is that the cloud ice content is too low in midlatitude continental convection without convective microphysics parameterization (Song et al., 2012), which cannot be improved by merely increasing the NBL.

Other parameters that we tested in the ZM cumulus parameterization with the dilute CAPE calculation include convective entrainment rate (zmconv_dmpdz), the convection adjustment time scale (zmconv_tau), the coefficient of autoconversion rate

(zmconv_c0_lnd), ice particle size (clubb_ice_deep), convective fraction (cldfrc_dp), and number of layers allowed for negative CAPE (zmconv_cape_cin). The overall conclusion is that separately tuning any of these parameters does not improve the simulation of echo top height. For the convective entrainment rate (zmconv_dmpdz), we decreased its value from $-0.7 \times 10^{-3}$ to $-1.0 \times 10^{-5}$, which means that the entrainment in convection is almost turned off, similar to the undiluted CAPE assumption. Results show the simulated echo top height is increased by 500-800 m in the EAMv1-test simulation, and the reflectivity span

in the lower troposphere is narrowed by 1-2 dBZ, which is closer to the observations (Fig. 8). This result is consistent with the

previous studies that tested the undiluted CAPE assumption as well (Neale et al., 2008; Hannah and Maloney, 2014). However, that assumption is unrealistic given the fact that the undiluted CAPE-based closure strongly deviated from observations (Zhang, 2009). In summary, changing any of our selected parameters individually in the ZM scheme does not improve the simulation of echo top height.

The MG2 cloud microphysics parameterization in E3SM determines only large-scale cloud and precipitation (i.e., those resolved by the model). Changes in the MG2 cloud microphysics parameterization could affect the parameterized cumulus cloud and precipitation by changing the large-scale forcing which feeds into the cumulus cloud calculations. By decreasing the MG2 autoconversion rate (prc_coef1), ideally the depletion of moisture within the atmospheric column is slowed down and more water vapor can be supplied to cumulus convection. Results show, however, that the echo top height is not affected

by changing the MG2 assumptions. Attempts at accelerating the Wegener–Bergeron–Findeisen process in MG2 to increase the conversion of liquid to snow/ice, as well as using lower size threshold for the ice-to-snow conversion have also proven to be unimportant to the simulation of echo top height.

    Thus, echo top height proves to be insensitive to the available tunable parameters. Setting the value of convective entrainment rate to be unrealistically low only gains a 500-800 m increment in echo top height. Given that the model underestimation is

more than 5 km, the increment is insufficient to solve the discrepancy. Note that each individual tunable parameter was changed without retuning the model to keep the top-of-atmosphere radiative energy budget balanced and the model performance optimized. Thus, some expected improvement in echo top height can be subsequently offset by other untuned processes. Instead of providing quantification of how the model responds to the changes of parameters, we emphasize the trend of change in echo top height, in which the simulation of the echo top height cannot be significantly improved by tuning only one of those

physical parameters. Further investigation of combinations of two and more parameters is a topic for a future study.

**4 Conclusions and Discussion**

We have evaluated the model performance of E3SM EAMv1 in simulating the warm-season 3D radar reflectivity at an hourly scale over the North American sector of the globe by comparing the model results to the 3D distribution of radar reflectivity observed by NEXRAD radars over the CONUS during April-September of 2014-2016. The evaluation is achieved by

improving the COSP radar simulator and employing special data processing techniques to ensure fair comparison between model and observations that are different in sampling frequency, horizontal-vertical resolutions, and minimum detection limit. We find that:

    1.    With the default microphysics assumptions in COSP, the simulated subgrid reflectivity PDF is bimodal, in disagreement with radar observations which show that the subgrid reflectivity follows a gamma distribution.

Changing the microphysics assumptions in COSP to be consistent with the MG2 microphysics parameterization used in E3SM, the bimodality of the subgrid distribution is nearly eliminated. It is therefore important to maintain consistency of microphysics assumptions between the host model and radar-echo simulator attached to the model as

advocated by the COSP community (Swales et al. 2018). For more accurate model evaluation, a higher-order consistency between the COSP and the host model is warranted in future studies.

2.   Below the 4-km altitude, the simulated domain-mean reflectivities by EAMv1 agree with NEXRAD observations in the magnitude, but the simulation fails to capture the spatial variability. The model underestimates the reflectivity in central U.S. between the Rocky Mountains and Mississippi River. This pattern suggests that the model is not adequately representing the mesoscale convective systems that dominate warm season rainfall in that region. The model overestimates the reflectivity outside this region.

3.   Above 4-km altitude, EAMv1 shows a severe underestimation of the domain-mean reflectivity, and the negative bias increases with altitude up to 11 km, above which model fails to simulate any valid reflectivity at all, whereas NEXRAD observations show strong radar echoes up to 16 km.

4.   EAMv1 is able to simulate the variability and extreme value of reflectivity at the lower troposphere but significantly underestimate them at high levels.

The NEXRAD observations used in this study reveal that EAMv1 fails to simulate the occurrence of large ice-phase particles at high levels in deep convective clouds. In addition, the conclusion of "simulated deep convection is not deep enough" also echoes the dry bias seen in GCMs as manifested in underestimations of total precipitation and individually large rain rates over the CONUS (e.g., Zheng et al., 2019). We have now shown that this model deficiency cannot be significantly improved by tuning a single value of the physical parameters in the ZM cumulus and MG2 cloud microphysics schemes. Note the large-
scale circulation is nudged towards observations for the simulations in this study, so our results represent the best-case model performance. Compared to the nudged simulations, free running of EAMv1 has shown nonnegligible biases in the regional circulation (Sun et al., 2019). With the nudged simulations, the large biases in circulation can be excluded so that the performances of physics parameterizations in simulating convective systems can be more insightfully understood.

The data processing techniques and metrics we have developed in this study can be used globally for model evaluation when
satellite-based radars provide global 3D radar observations. The GPM radar observations will eventually be able to provide global radar echo coverage (Houze et al., 2019), whose data have been proven consistent with NEXRAD (Wang et al., 2019b). However, as discussed in Section 2, the sampling by GPM at 1° model grid elements for only three years of GPM data is insufficient for obtaining robust grid-mean values to compare with the EAMv1 simulation. In addition to the restriction in the availability of observational data, the high computation cost with the incorporation of COSP simulator in simulation and the
demand of large data space (14,000 core hours and 1.2 TB data per simulation month at hourly output frequency) have hindered the modelling for an extended period. When GPM has run for a much longer time period and more powerful computational resources become available, it will be a very useful study to evaluate the long-term model simulations at the global scale. In addition, the results of this study can provide metrics for evaluating the cumulus parameterizations or provide insights for further improving the cumulus parameterizations like Labbouz et al. (2018), which can be a follow-on work. Future studies
can also focus on separately evaluating properties in convective and stratiform regions, since the thermodynamic and reflectivity profiles are fundamentally different between the two regions.

## Code Availability

The source code in this study is based on the Department of Energy (DOE) Energy Exascale Earth System Model (E3SM)
Project version 1 at revision 9a86ab9 whose code can be acquired from the E3SM repository (https://github.com/E3SM-Project/E3SM/tree/kaizhangpnl/atm/cm20170220), which is also permanently archived in the National Energy Research Scientific Computing Canter (NERSC) High Performance Storage System (HPSS) at https://portal.nersc.gov/archive/home/w/wang406/www/Publication/Wang2020GMD.

## Data Availability

The observational data is available through National Center for Atmospheric Research (NCAR) Research Data Archive (https://doi.org/10.5065/D6NK3CR7). Model results can be accessed from https://portal.nersc.gov/archive/home/w/wang406/www/Publication/Wang2020GMD.

## Author Contributions

Jingyu Wang performed the simulations and conducted the analyses. Jiwen Fan and Robert A. Houze Jr developed the idea of
this research. Kai Zhang helped on the model configuration and Po-Lun Ma implemented the radar simulator. Guang J. Zhang provided feedback and helped shape the research. All authors discussed the results and contributed to the final manuscript.

## Acknowledgement

We acknowledge the support of the Climate Model Development and Validation (CMDV) project at PNNL. The effort of J. Wang, J. Fan, Kai Zhang, and Po-Lun Ma was supported by CMDV. Robert A. Houze was supported by NASA Award
NNX16AD75G and by master agreement 243766 between the University of Washington and PNNL. Stella R. Brodzik was supported by NASA Award NNX16AD75G and subcontracts from the CMDV and Water Cycle and Climate Extreme Modeling (WACCEM) projects of PNNL. Guang J. Zhang was supported by the DOE Biological and Environmental Research Program (BER) Award DE-SC0019373. PNNL is operated for the US Department of Energy (DOE) by Battelle Memorial Institute under Contract DE-AC05-76RL01830. This research used resources of the National Energy Research Scientific
Computing Center (NERSC), a U.S. Department of Energy Office of Science User Facility operated under contract DE-AC02-05CH11231. The GridRad radar dataset is obtained at the Research Data Archive of the National Center for Atmospheric Research (NCAR) (https://rda.ucar.edu/datasets/ds841.0/).

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

**Table List**

Table 1. Modification of the hydrometeor assumptions used in COSP.

| Hydrometeor Type[1] | Distribution Type | | Density (kg m$^{-3}$) | | Particle Mean Diameter (μm) | | Distribution Width[2] (Unitless) | |
|---|---|---|---|---|---|---|---|---|
| | Default | Modified | Default | Modified | Default | Modified | Default | Modified |
| LSL | Lognormal | Gamma | 524×D$^3$ | - | 6 | 12 | 0.3 | 0 |
| CVL | Lognormal | Gamma | 524×D$^3$ | - | 6 | 12 | 0.3 | 0 |
| LSI | Gamma | - | 110.8×D$^{2.91}$ | 500 | 4 | - | 2 | 0 |
| CVI | Gamma | - | 110.8×D$^{2.91}$ | 500 | 4 | - | 2 | 0 |
| LSS | Exponential | - | 100 | 250 | N/A | - | N/A | - |
| CVS | Exponential | - | 100 | 250 | N/A | - | N/A | - |

[1]LS: Large-Scale; CV: Convective; L: Cloud Liquid; I: Cloud Ice; S: Snow.

[2]Distribution width: $v$ in $N(D) = N_0 D^{(v-1)}e^{-\lambda D}$, which is a shape parameter in Gamma distribution describing the dispersion of the distribution.

Table 2. The statistical comparison of radar reflectivity between NEXRAD and EAMv1

| Altitude | NEXRAD | | | | EAMv1 | | | |
|---|---|---|---|---|---|---|---|---|
| | Mean (dBZ) | Standard Deviation (dBZ) | 95th Percentile (dBZ) | Sample Numbers | Mean (dBZ) | Standard Deviation (dBZ) | 95th Percentile (dBZ) | Sample Numbers |
| 2 km | 25.1 | 7.7 | 32.1 | $1.7 \times 10^6$ | 28.7 | 7.4 | 35.8 | $4.1 \times 10^6$ |
| 4 km | 24.0 | 7.2 | 31.6 | $1.6 \times 10^6$ | 24.0 | 6.4 | 30.2 | $4.2 \times 10^6$ |
| 8 km | 19.2 | 5.2 | 24.4 | $7.9 \times 10^5$ | 15.0 | 3.9 | 21.0 | $1.5 \times 10^6$ |
| 11 km | 16.6 | 4.4 | 21.8 | $2.2 \times 10^5$ | 9.8 | 1.6 | 12.9 | $4.1 \times 10^3$ |






Table 3. Changes of the tunable parameters in the sensitivity tests for echo top height.

| | Parameter | Physics Meaning | Default | Changed Values | Impact |
|---|---|---|---|---|---|
| **Cumulus parameterization** | NBL restriction | The upper limit level of the integral of the mass flux, moist static energy etc. in ZM | Calculated NBL | 200 hPa, 70 hPa | No |
| | zmconv_dmpdz | ZM entrainment rate in CAPE calculation | -0.7e-3 | -1.0e-3, -1.0e-5 | Yes |
| | zmconv_tau | Convection adjustment time scale | 1 hr | 15min, 6 hr | No |
| | zmconv_c0_lnd | Coefficient of autoconversion rate in ZM | 0.007 | 0.01, 0.002 | No |
| | zmconv_cape_cin | Number of layers allowed for negative CAPE | 1 | 5, 10 | No |
| | clubb_ice_deep | Assumed ice condensate radius detrained from ZM | 16e-6 | 32e-6, 8e-6 | No |
| | cldfrc_dp1 | Convective fraction | 0.045 | 0.01, 0.2 | No |
| **Microphysics parameterization** | prc_coef1 | Coefficient of autoconversion rate in MG2 | 30500 | 10000, 675 | No |
| | berg_eff_factor | Efficiency factor for the Wegener–Bergeron–Findeisen process | 0.1 | 0.2, 0.7 | No |
| | thres_ice_snow | Autoconversion size threshold from cloud ice to snow | Temperature dependent | Maximize at 175e-6 | No |


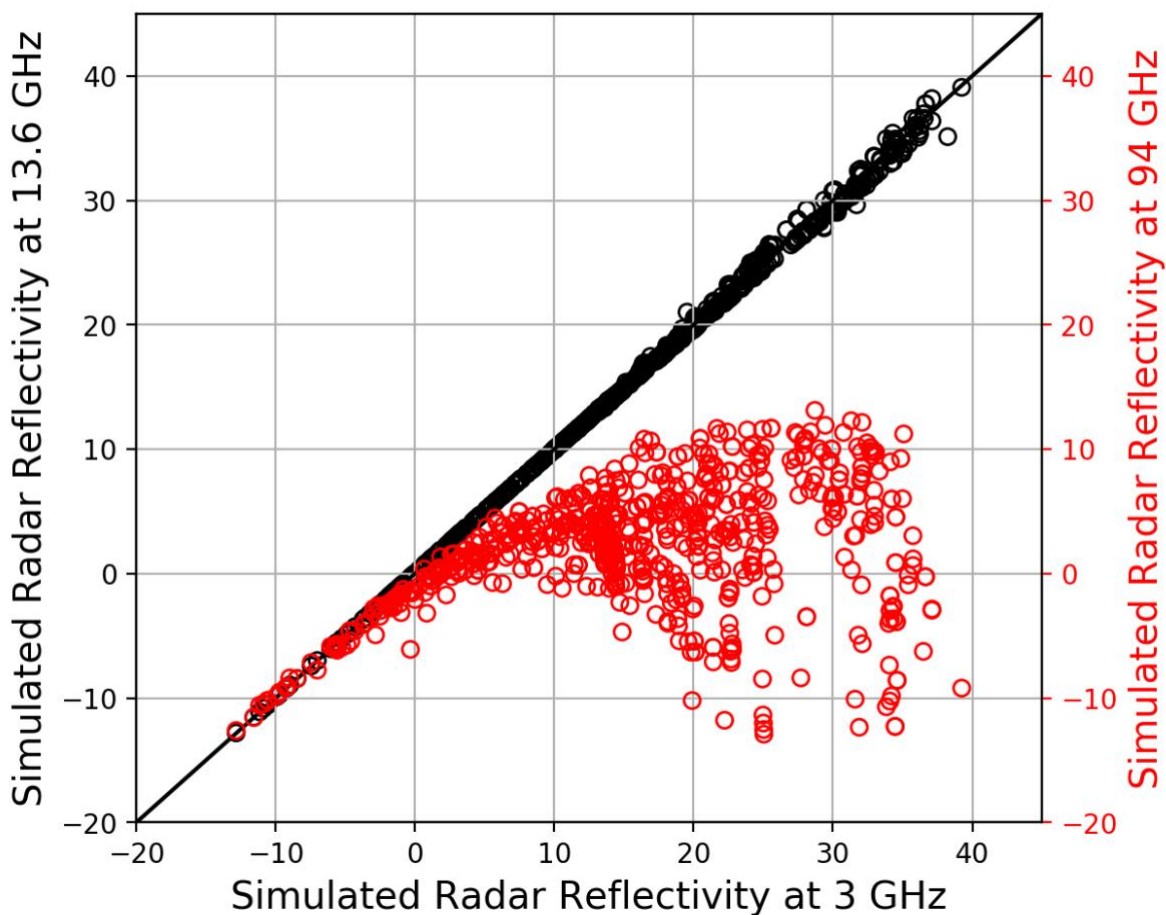

**Figure 1: Scatter plots between radar reflectivity values simulated by the COSP simulator at 3 GHz (x-axis) versus those simulated at 13.6 GHz (left y-axis) and 94 GHz (right y-axis).**



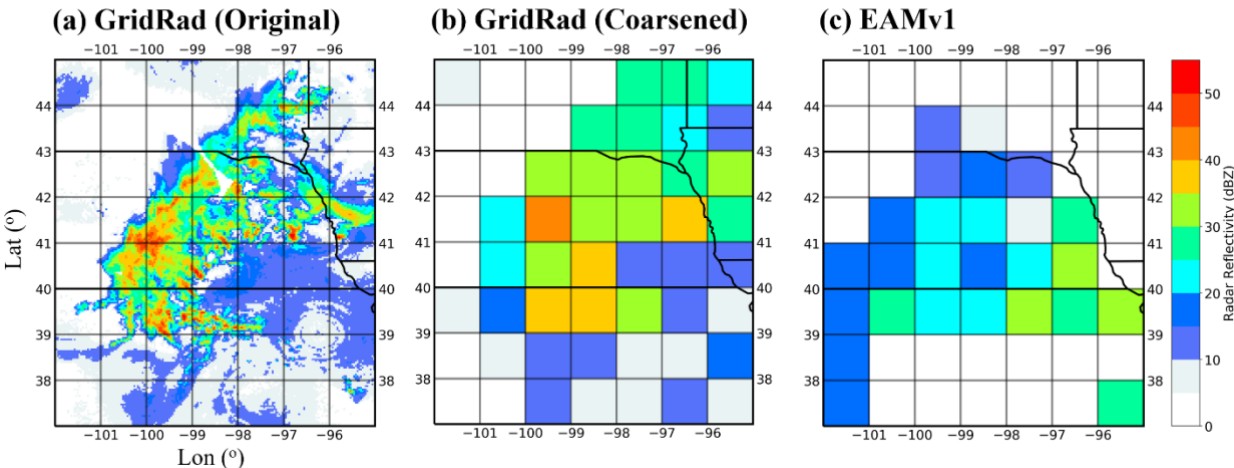

**Figure 2: Examples of (a) original GridRad observation, (b) GridRad mapped over the E3SM model grid, and (c) the concurrent model simulation on 2016 May 11, 07:00 UTC, at the 2-km altitude.**


# The Comparison of Radar Reflectivity Subgrid Distribution

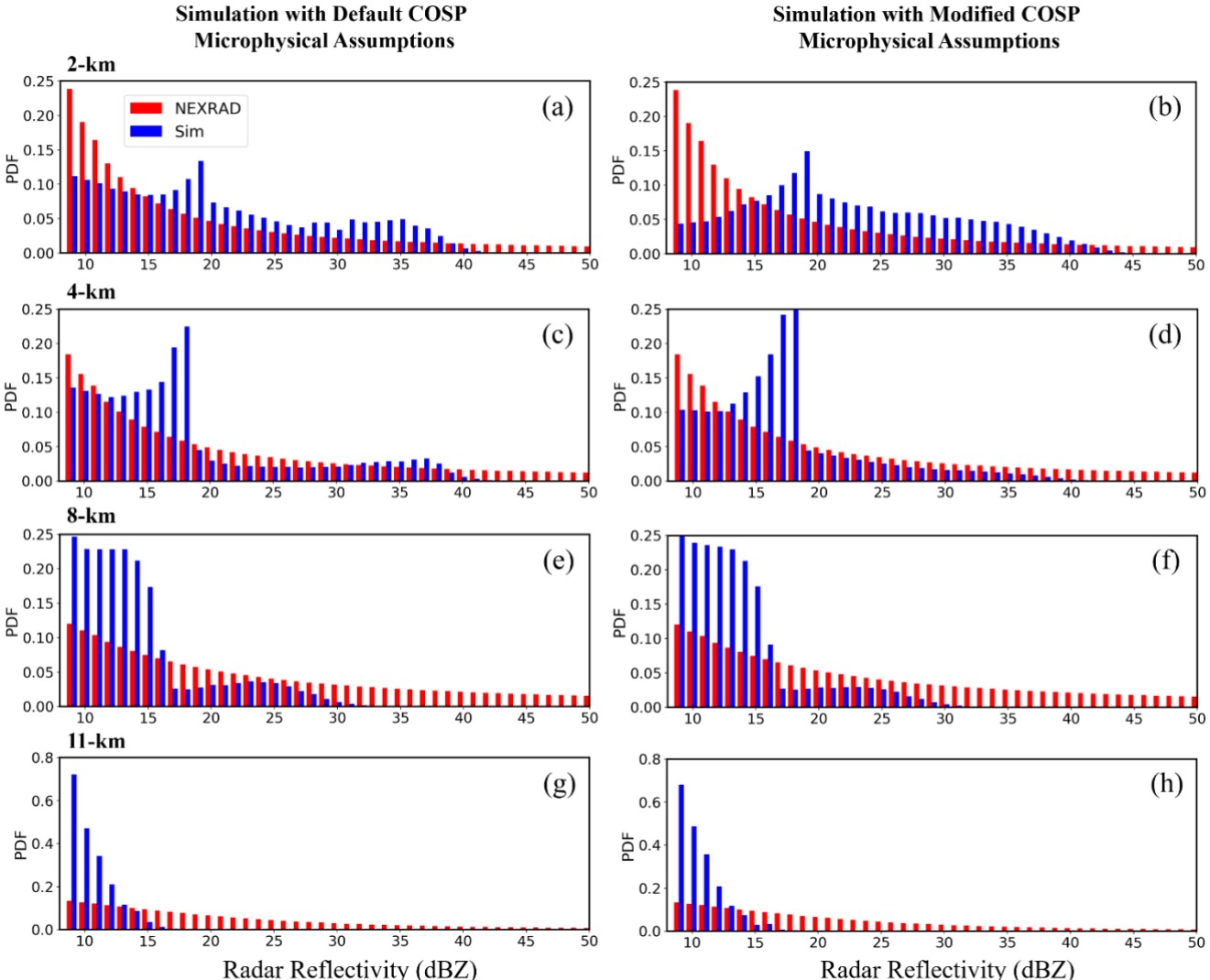

**Figure 3: Comparison of radar reflectivity subgrid distribution between NEXRAD observations (red bars) and the simulations (blue bars) at the vertical levels of 2 km, 4 km, 8 km, and 11 km. Simulation results in the left and right columns are from the default microphysics assumptions in COSP and modified COSP microphysics assumptions, respectively.**

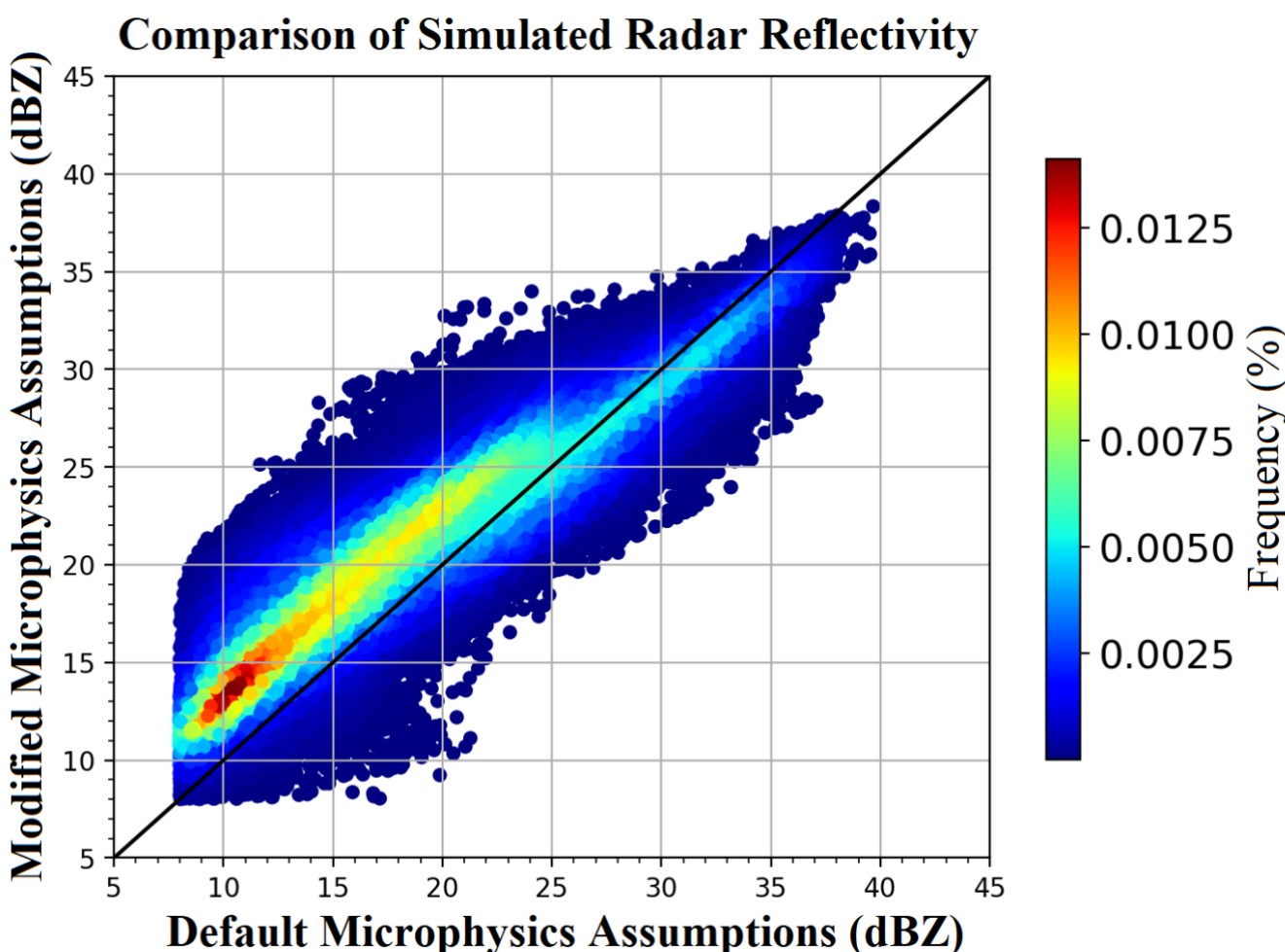

**Figure 4: Scatter density plot between radar reflectivity values from the simulation with the modified microphysics assumptions (y-axis) versus those with the default microphysics assumptions (x-axis). The data shown are for April 2014. The dots are color labelled with their frequency of occurrence.**



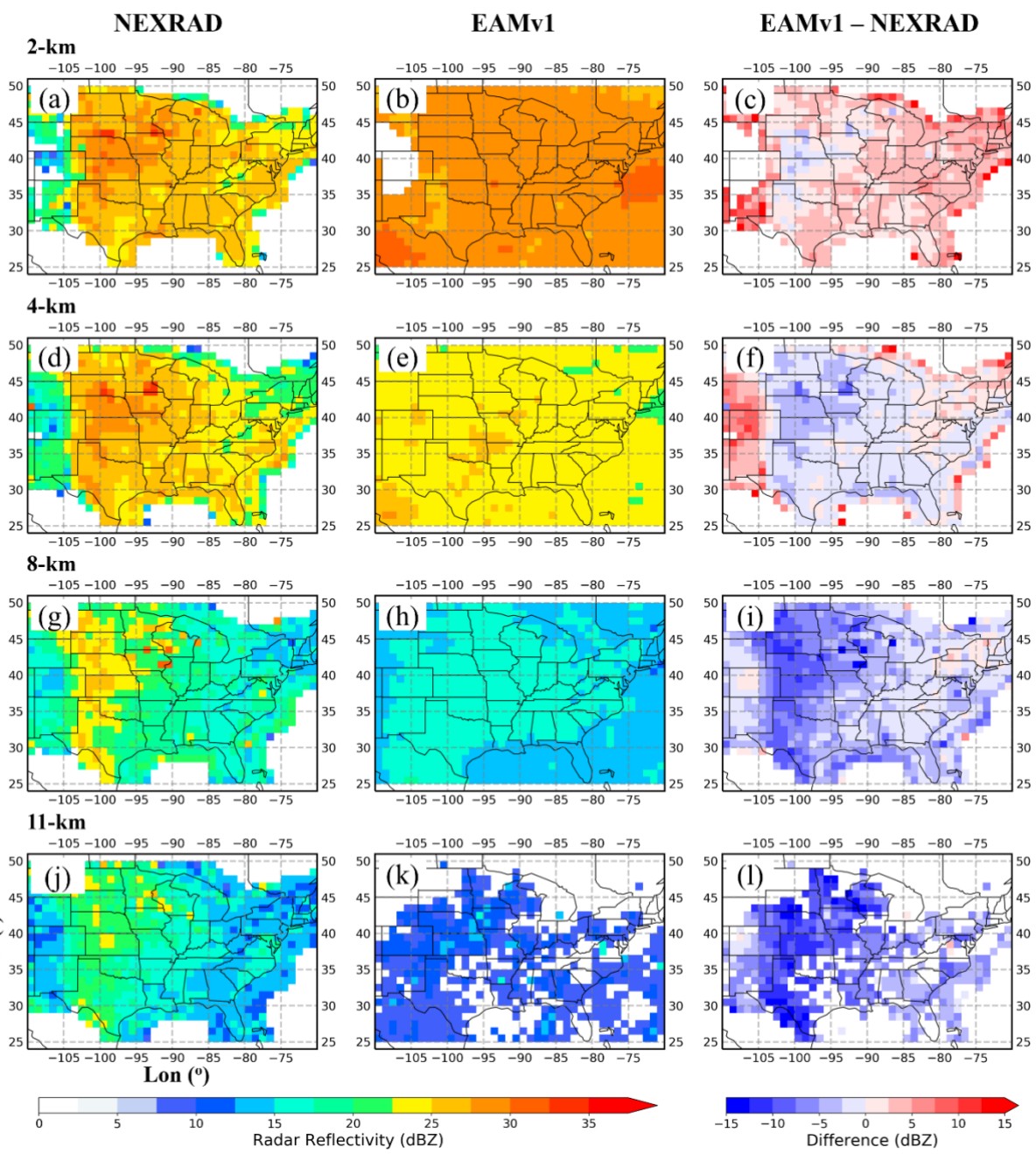

Figure 5: Plan view of radar reflectivity averaged from NEXRAD observations (a, d, g, j), EAMv1 simulation with the modified microphysics assumptions in COSP (b, e, h, k), as well as their absolute differences (c, f, i, l) at the level of 2-km, 4-km, 8-km, and

**11-km altitude. The NEXRAD data are spatially averaged from native resolution to the model grid over 2014-2016 April-September period, and the simulation are vertically interpolated to the NEXRAD levels.**


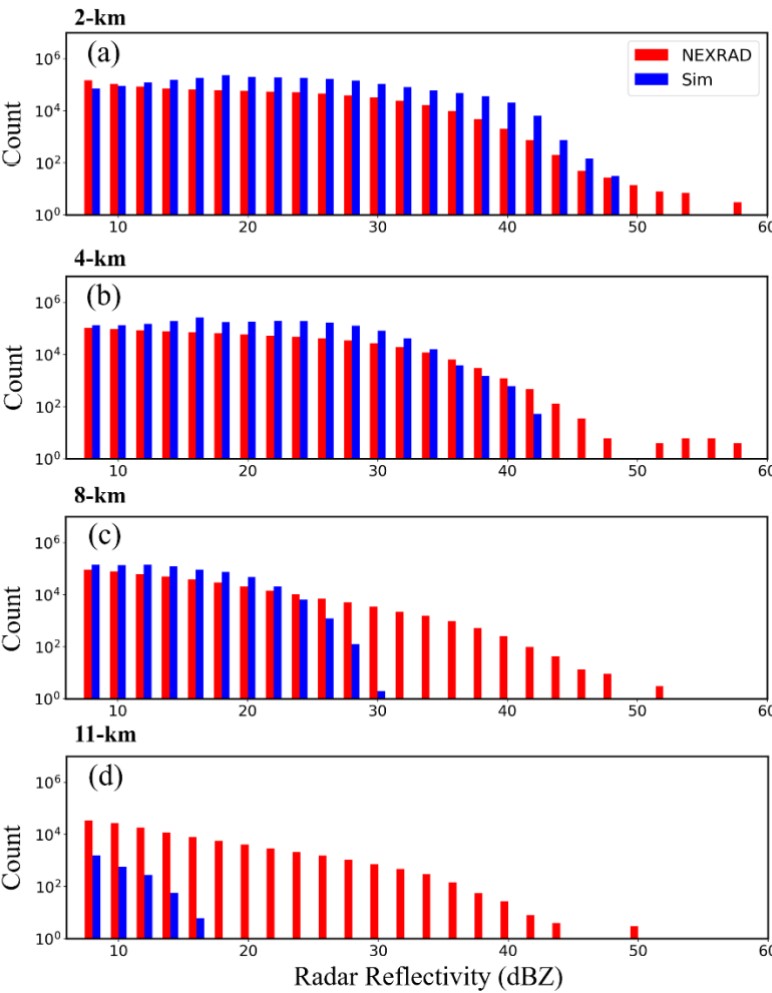

**Figure 6: Comparison of radar reflectivity histograms at 1° scale between NEXRAD observations (red bars) and the simulations (blue bars) at the vertical levels of 2 km, 4 km, 8 km, and 11 km.**

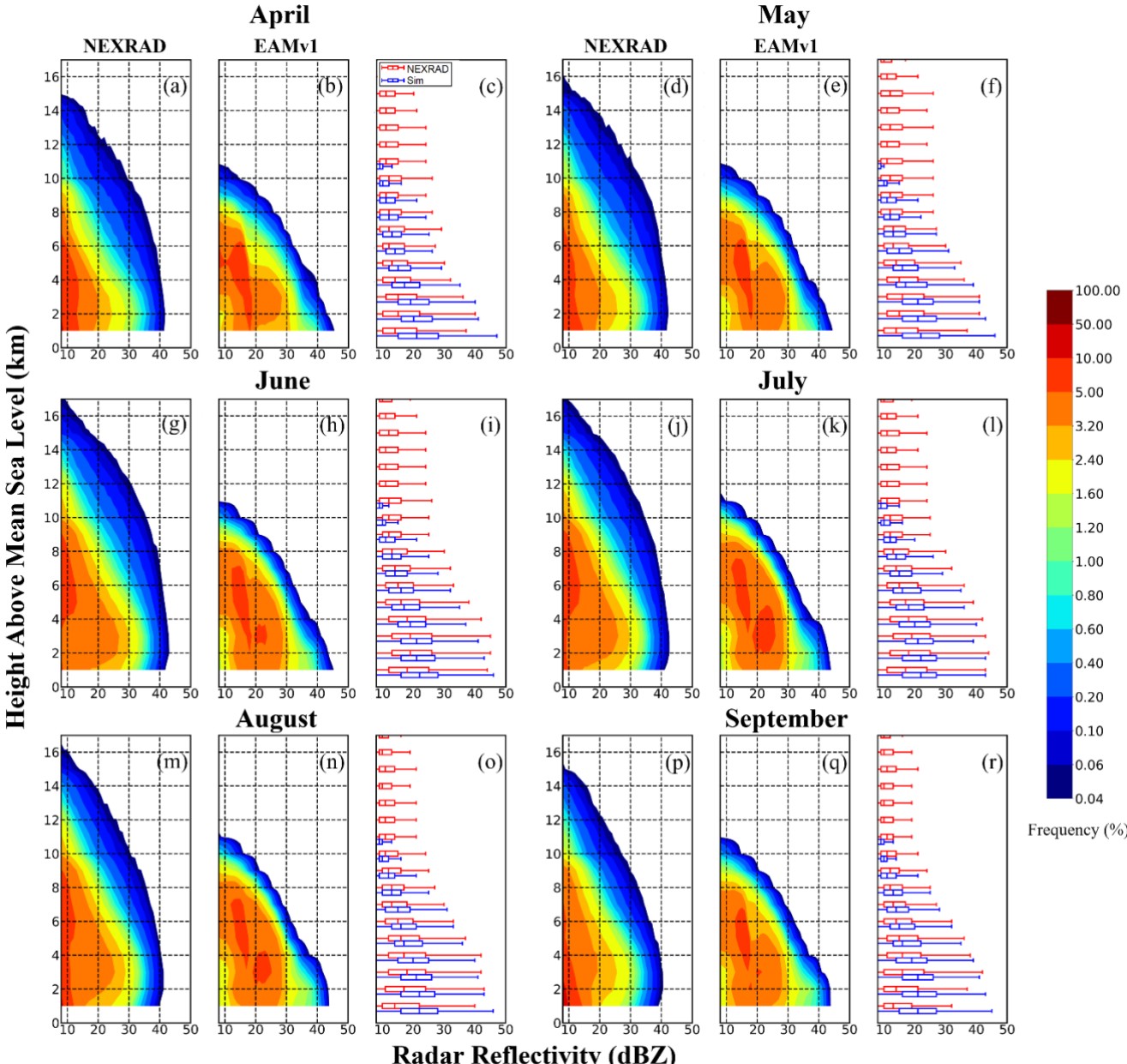

Figure 7: Contoured-Frequency-by-Altitude-Diagrams (CFADs) normalized by the total number of samples at all altitude levels for NEXRAD (a, d, g, j, m, p) and EAMv1 simulation with the modified microphysics assumptions in COSP (b, e, h, k, n, q) for the months from April to September averaged over 2014-2016 period. The box-whisker plots (c, f, i, l, o, r) for NEXRAD (red) and EAMv1(blue) are calculated using normalization at each individual level, where the center of the box represents the 50th percentile value, and the 25th and 75th percentiles are represented by the left and right boundary of the box, respectively. Whiskers correspond to the 5% and 95% values.

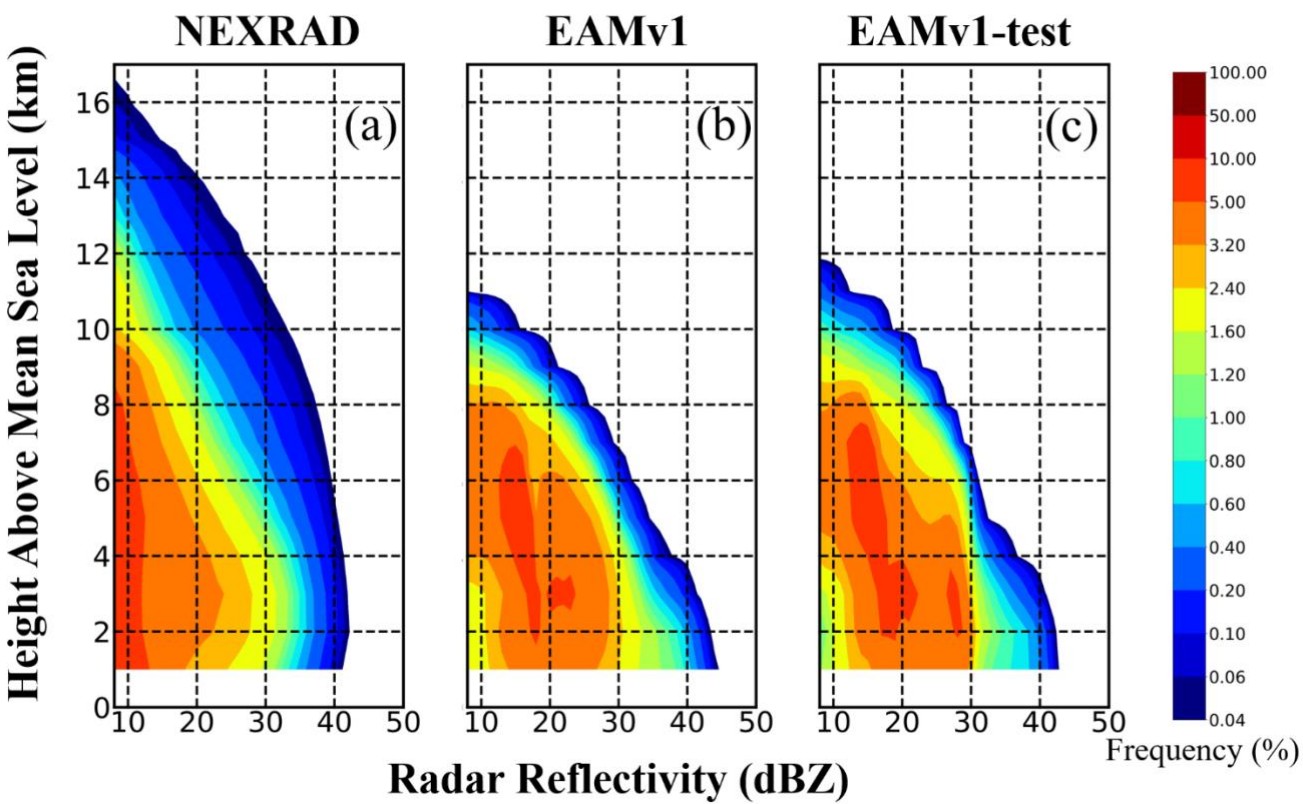

**Figure 8:** Comparison of Contoured-Frequency-by-Altitude-Diagrams (CFADs) for the warm seasons over 2014-2016 between (a) NEXRAD, (b) EAMv1 simulation, and (c) the EAMv1-test simulation with reduced convective entrainment rate.