# Peer review of "Using Radar Observations to Evaluate 3D Radar Echo Structure Simulated by the Global Model E3SM Version 1"

_Geoscientific Model Development, 2020_

## Short Comment (SC1) · 25 May 2020

Dear authors,

in my role as Executive editor of GMD, I would like to bring to your attention our Editorial version 1.2:

https://www.geosci-model-dev.net/12/2215/2019/

This highlights some requirements of papers published in GMD, which is also available on the GMD website in the 'Manuscript Types' section:

http://www.geoscientific-model-development.net/submission/manuscript_types.html

[Figure]

In particular, please note that for your paper, the following requirements have not been met in the Discussions paper:

- "If the model development relates to a single model then the model name and the version number must be included in the title of the paper. If the main intention of an article is to make a general (i.e. model independent) statement about the usefulness of a new development, but the usefulness is shown with the help of one specific model, the model name and version number must be stated in the title. The title could have a form such as, "Title outlining amazing generic advance: a case study with Model XXX (version Y)"."

- "Code must be published on a persistent public archive with a unique identifier for the exact model version described in the paper or uploaded to the supplement, unless this is impossible for reasons beyond the control of authors. All papers must include a section, at the end of the paper, entitled "Code availability". Here, either instructions for obtaining the code, or the reasons why the code is not available should be clearly stated. It is preferred for the code to be uploaded as a supplement or to be made available at a data repository with an associated DOI (digital object identifier) for the exact model version described in the paper. Alternatively, for established models, there may be an existing means of accessing the code through a particular system. In this case, there must exist a means of permanently accessing the precise model version described in the paper. In some cases, authors may prefer to put models on their own website, or to act as a point of contact for obtaining the code. Given the impermanence of websites and email addresses, this is not encouraged, and authors should consider improving the availability with a more permanent arrangement. Making code available through personal websites or via email contact to the authors is not sufficient. After the paper is accepted the model archive should be updated to include a link to the GMD paper."

As the only model used in your study is the E3SM model, provide its name and version number in the title. (e.g., "by the global mode E3SM (V...)"

Additionally, note that Github is not a permanent repository. Thus ensure permanent archiving of the exact version used for this publication. (e.g., by upload on Zenodo).

Best regards, Astrid Kerkweg (Executive Editor)

---

## Referee Comment (RC1) · Anonymous Referee #1 · 22 Jun 2020

This is a nice paper testing the performance of the NCAR climate model against observations from the US weather radar network. My view is this is important and that more climate models need to be tested against observations, both for current and recent climate but also studies such as this focusing on representation of key processes.

Note the large scale circulation in this model is being nudged towards observations, so that the performance being discussed represents an upper bound. It would be interesting to also compare the output from an extended period of the model in free running mode as for CMIP and looking at the latter part of 20th Century and early 21st century runs so that the forcings are consistent with current observations. I suspect the

key limitations outlined in the nudged runs will be at least as large and possibly greater.

The analysis looks at a few metrics including the overall spatial distribution and the vertical profiles of reflectivity. This is OK as far as it goes, but I feel further and deeper analysis will yield more information on the process limitations in the model. For example, further insights would be gained by examining the mean diurnal cycle of convection and how that compares with observations over the great Plains. Does the model reproduce the night-time maxima over the eastern plains and as a difficult test are propagating modes observed modulating the diurnal convective activity (cf. Carbone and Tuttle, J Clim., 2008). Is the spatial distribution of time of peak convection at all captured or is it dominated by a morning maxima as convection triggers too early in daily heating as occurs in many simulations in the tropics. The diurnal cycle of convection is also important for the resulting cloud and radiation climatology of the model.

In Sect 3.1, where there is a mean difference in reflectivity. Noting these are linear averages over a 100 km area, do you have a feel how much of this is associated with differing convective fractions within the grid points, differing fractions of precipitation or differences in the PDF of the reflectivities not associated with the convective/precipitating fraction? Have you compared convective fraction from the model paramaterisations with observations? Diagnostics looking at fractional cover can also aid interpretation and diagnose issues.

In Section 3.3, comparing NEXRAD and subscale distributions – testing maybe could be earlier in the paper and is there is a degree of circularity in your argument since you are adjusting the sub-grid scale distributions with the observed NEXRAD data and so naturally there is increased agreement. Note that the bimodality in the original distributions shown in Fig 4 are not generally observed in nature.

As a minor point, on line L130, using NEXRAD also simplifies the radar scattering calculations compared with GPM and TRMM with the 10 cm wavelength radar being close to Rayleigh scatter most of the time although the scattering calculations are still

complex for ice habits

Overall, I think this is a useful study, but would benefit from being taken further. It is clearly addressing important issues with climate models and as noted these kind of studies are sorely needed. The methods are clearly articulated.

---

## Referee Comment (RC2) · Alain Protat (Referee) · 25 Jun 2020

This paper uses three years of post-processed NEXRAD radar data to assess the performance of a recent version of the new E3SM model, using reflectivity simulations. The results are interesting, and the presentation of the results is clear. I have a number of comments and suggestions to improve the scientific content of the paper, which is a bit on the light side. They are all somewhat minor comments for consideration by the authors, except one more major comment that should be fully addressed. Therefore, I recommend the paper be accepted with minor revisions provided that at least my major comment is addressed.

**Major comment:**

My main comment is about the use of an 8 dBZ threshold and the implication for echo top height conclusions in the paper. There is nothing wrong with using such threshold, but it introduces in my opinion a possible misinterpretation of the results regarding the echo top height statistics to address the all-important question: does my model reproduces the vertical development of convection well, statistically? Comparing echo top heights between observations and models is very tricky, because using a threshold in reflectivity implicitly carries the assumption that the echo top height is not affected by the threshold when you draw conclusions. Let me give an example: say the model is underestimating reflectivities in ice phase by 5 dBZ (in your case it seems to be more than that). If you want to learn something about how good the model approximates cloud top height statistics (and indirectly your vertical air velocities in deep convection), you should actually use 3 dB echo top heights from the model compared to 8 dB echo top height in the observations to be fair to the model. In your case, you find a substantial underestimation of reflectivities in the upper levels, well, then it's not surprising that you are underestimating the 8 dB echo top height by a large amount. But what do you learn with this about the deficiencies in the model, especially about the convective vertical velocities and convective mass flux assumptions. On another hand if you take the model cloud top height, chances are that the NEXRAD radars won't have the sensitivity to detect it, and this time the radar statistics will show lower cloud top heights than the model (I think I have seen statistics somewhere showing a systematic ~ 2km difference between cloud top height and 0-dBZ echo top heights, but I can't find the reference just yet). If it does not though, it would mean that the model really underestimates cloud top heghts, which would be very interesting. You could check, maybe.

There is a very good discussion on this in the Labbouz et al. (2018) paper. This paper also tackles similar issues but using different types of comparisons, so I believe it should be quoted in your paper (as well as references therein).

**Specific comments:**

1. The 13.6 GHz versus 3 GHz difference: my only issue with this is that you should make sure that you have switched off any Ku-band attenuation correction in COSP. Have you? If so you should mention it in the paper.

2. Line 139: it does not make sense to say that the minimum detectable reflectivity of NEXRAD radars is 0 dBZ. The MDR varies with range (MDR(range)=MDR(1km) + 20 log10(range_km)), so you either need to provide the sensitivity at 1 or 10 or 100km (whatever you prefer) or state that a hard threshold of 0 dBZ is applied somewhere in the NEXRAD processing. Also, I would be surprised if the NEXRAD radars can detect 0 dBZ at 250km range (but I don't have those numbers).

3. Back to the 8-dBZ threshold: how sensitive are all your results to the use of 8-dB threshold? What happens if you take 0 dB (you say NEXRAD can detect 0dB), so why didn't you use 0 dB to get closer to the true cloud top? It would make more sense in my opinion.

4. Line 180-181: again, you say here that your most distinct result is about echo top height. But I believe it does not teach you anything about potential model deficiencies, including the convective vertical velocities and mass flux issues highlighted in Labbouz et al. (2018) and others. Your main result is actually that the model is underestimating reflectivities, but may well be excellent at producing realistic cloud tops. The only thing is that they have lower reflectivities than in reality.

5. Line 194: this statement is likely wrong. I would bet that it simulates reflectivities at heights greater than 11km, but they are under your 8 dB threshold due to ice microphysics deficiencies.

6. Line 226: ~5dBZ: it is more like 4 dB and only for Z<25 dBZ, so maybe this statement should be modified to reflect this.

7. Line 323: typo "Brodzik"

8. Figure 2 and associated discussion in section 3: comparing mean reflectivities is interesting, but only one aspect of what you'd like to get right with a model. The two main other things I would personally try to assess is the standard deviation of reflectivity at 100km scale (to check if the model reproduces the observed variability even if it does not have the mean right) and the 95[th] percentile or even 99[th] percentile if you have enough samples (to check if the model has any skills in forecasting extremes). These two additional things would greatly enhance the scientific content and scope of this model evaluation exercise.

References:

*Labbouz, L., Z. Kipling, P. Stier, and A. Protat, 2018: How well can we represent the spectrum of convective clouds in a climate model? Journal of the Atmospheric Sciences, 75(5), 1509–1524.*

Good luck with the review,

Alain Protat

Melbourne, 25/06/2020

---

## Author Comment (AC1) · 13 Aug 2020

**Response to Editor**

Thank you very much for the advice to improve the quality of this manuscript. We carefully addressed the comments and made corresponding changes to the manuscript. In this "response to editor" document, we provided detailed responses in **blue bold as below**.

Editor's comments

As the only model used in your study is the E3SM model, provide its name and version number in the title. (e.g., "by the global mode E3SM (V...)"

**The title of this manuscript has been changed to "Using Radar Observations to Evaluate 3D Radar Echo Structure Simulated by the Global Model E3SM Version 1".**

Additionally, note that Github is not a permanent repository. Thus ensure permanent archiving of the exact version used for this publication. (e.g., by upload on Zenodo).

**The model source code has been permanently archived in the National Energy Research Scientific Computing Canter (NERSC) High Performance Storage System (HPSS) at https://portal.nersc.gov/archive/home/w/wang406/www/Publication/Wang2020GMD. This has been added to the code and data availability sections in the revised manuscript.**

---

## Author Comment (AC2) · 13 Aug 2020

**Responses to Reviewer #1**

This is a nice paper testing the performance of the NCAR climate model against observations from the US weather radar network. My view is this is important and that more climate models need to be tested against observations, both for current and recent climate but also studies such as this focusing on representation of key processes.

**We thank the reviewer for recognizing our efforts and providing helpful suggestions and comments. Our point-by-point responses are provided as below.**

Note the large scale circulation in this model is being nudged towards observations, so that the performance being discussed represents an upper bound. It would be interesting to also compare the output from an extended period of the model in free running mode as for CMIP and looking at the latter part of 20th Century and early 21st century runs so that the forcings are consistent with current observations. I suspect the key limitations outlined in the nudged runs will be at least as large and possibly greater.

**Free-run simulations could cause a large bias in circulation, especially regional circulation, making the comparison with observation in convective systems difficult. We wanted to start from the nudged runs to exclude the large biases from circulation. We agree that further study can be extended to free runs. We have added this clarification in the conclusions: "Note the large-scale circulation is nudged towards observations for the simulations in this study, which represents the upper bound of model performance. Compared to the nudged simulations, the free running of EAMv1 has shown nonnegligible biases in the regional circulation (Sun et al., 2019). With the nudged simulations, the large biases in circulation can be excluded so that the performances of physics parameterizations in simulating convective systems can be more insightfully understood." We were not able to provide evaluation for a longer period, because "in addition to the restriction in the availability of observational data, the high computation cost with the incorporation of COSP simulator in simulation and the demand of large data space (14,000 core hours and 1.2 TB data per simulation month at hourly output frequency) have hindered the modeling for an extended period."**

The analysis looks at a few metrics including the overall spatial distribution and the vertical profiles of reflectivity. This is OK as far as it goes, but I feel further and deeper analysis will yield more information on the process limitations in the model. For example, further insights would be gained by examining the mean diurnal cycle of convection and how that compares with observations over the great Plains. Does the model reproduce the night-time maxima over the eastern plains and as a difficult test are propagating modes observed modulating the diurnal convective activity (cf. Carbone and Tuttle, J Clim., 2008). Is the spatial distribution of time of peak convection at all captured or is it dominated by a morning maxima as convection triggers too early in daily heating as occurs in many simulations in the tropics. The diurnal cycle of convection is also important for the resulting cloud and radiation climatology of the model.

**The precipitation including the diurnal cycle has been evaluated for EAMv1 (Zheng et al., 2019), which showed the model failed to simulate the diurnal variation of precipitation over the central United States. To avoid the redundancy, here we have added a plot for comparing the diurnal cycle of column-maximum reflectivity (Fig. 6), which can indicate the intensity of precipitation. The following text has been added to the last paragraph of Section 3.3, "As evaluated in Zheng et al. (2019), E3SM v1 failed to simulate the diurnal variation of precipitation over the central United States. Here we examine the diurnal cycle of column-maximum reflectivity (Fig. 6), which can**

**indicate the intensity of precipitation (Carbone and Tuttle, 2008). The observation shows two peaks, one in the early morning and the other in the late afternoon. This pattern differs from the observation of total precipitation, which only has one nocturnal peak with a smooth transition from the minimum at local noon. The difference between the two observed variables is expected, as the column-maximum reflectivity most likely represents convective (not stratiform) precipitation, which occurs significantly in the early morning and late afternoon. In contrast with the two peaks in observed column-maximum reflectivity, the EAMv1 simulation demonstrates a flat diurnal curve without any obvious peak, suggesting the model has a difficulty in simulating the convective precipitation.**

In Sect 3.1, where there is a mean difference in reflectivity. Noting these are linear averages over a 100 km area, do you have a feel how much of this is associated with differing convective fractions within the grid points, differing fractions of precipitation or differences in the PDF of the reflectivities not associated with the convective/precipitating fraction? Have you compared convective fraction from the model paramaterisations with observations? Diagnostics looking at fractional cover can also aid interpretation and diagnose issues.

**Thanks for the comment. We did not output the convective fraction for model grids, which prevents us from looking at the relationship of mean reflectivity with the convective fraction. Since the subgrid distributions of cloud and precipitation assumed in the COSP simulator have nothing to do with a convective fraction which is calculated by the ZM cumulus parameterization, we think the relationship of mean reflectivity with the convective fraction might not mean much.**

In Section 3.3, comparing NEXRAD and subscale distributions – testing maybe could be earlier in the paper and is there is a degree of circularity in your argument since you are adjusting the sub-grid scale distributions with the observed NEXRAD data and so naturally there is increased agreement. Note that the bimodality in the original distributions shown in Fig 4 are not generally observed in nature.

**We agree with the reviewer that the structure of the manuscript should be adjusted to avoid circularity. Section 3.3 has been moved to the beginning of the results.**

**We agree that the bimodality shown in original distribution in the left column of Figure 5 disagree with the general observation. However, this was the result from the COSP in the E3SM v1 in which cloud microphysical parameters are not aligned with the microphysics scheme used in the host model. After the correction, the distribution is more-like a Gamma distribution,**

As a minor point, on line L130, using NEXRAD also simplifies the radar scattering calculations compared with GPM and TRMM with the 10 cm wavelength radar being close to Rayleigh scatter most of the time although the scattering calculations are still complex for ice habits.

**We agree with the reviewer and have added a clarification at the end of Section 2.3, i.e., "For the NEXRAD observation, its 10 cm wavelength guarantees Rayleigh scattering for most situations."**

Overall, I think this is a useful study, but would benefit from being taken further. It is clearly addressing important issues with climate models and as noted these kind of studies are sorely needed. The methods are clearly articulated.

**Thanks.**

---

## Author Comment (AC3) · 13 Aug 2020

**Reponses to Reviewer #2**

This paper uses three years of post-processed NEXRAD radar data to assess the performance of a recent version of the new E3SM model, using reflectivity simulations. The results are interesting, and the presentation of the results is clear. I have a number of comments and suggestions to improve the scientific content of the paper, which is a bit on the light side. They are all somewhat minor comments for consideration by the authors, except one more major comment that should be fully addressed. Therefore, I recommend the paper be accepted with minor revisions provided that at least my major comment is addressed.

**Thanks Alain, for your valuable comments to improve the paper. See our point-by-point responses as below.**

**Major comment:**

My main comment is about the use of an 8 dBZ threshold and the implication for echo top height conclusions in the paper. There is nothing wrong with using such threshold, but it introduces in my opinion a possible misinterpretation of the results regarding the echo top height statistics to address the all-important question: does my model reproduces the vertical development of convection well, statistically? Comparing echo top heights between observations and models is very tricky, because using a threshold in reflectivity implicitly carries the assumption that the echo top height is not affected by the threshold when you draw conclusions. Let me give an example: say the model is underestimating reflectivities in ice phase by 5 dBZ (in your case it seems to be more than that). If you want to learn something about how good the model approximates cloud top height statistics (and indirectly your vertical air velocities in deep convection), you should actually use 3 dB echo top heights from the model compared to 8 dB echo top height in the observations to be fair to the model. In your case, you find a substantial underestimation of reflectivities in the upper levels, well, then it's not surprising that you are underestimating the 8 dB echo top height by a large amount.

**Thanks for the comment. We agree with the reviewer on the possible caveats with the threshold of reflectivity. But this cannot be avoided when comparing with model values over a 100-km grid. We have used a lower threshold of 0 dBZ to see how the results are sensitive to the choice of the threshold. As shown in Figure R1, we do see an increment of ~1 km in the simulated echo top height, however the observation doesn't change much. As a result, switching to the lower threshold of 0 dBZ has a very limited impact on the main conclusion that the model severely underestimates the echo top height.**

[Figure]

**Figure R1. The sensitivity test of changing the minimum reflectivity threshold from 8 dBZ (a, b) to 0 dBZ (c, d).**

We have added statements to Section 3.3, "From Fig. 5 it is clear that the model severely underestimates the echo top height by at least 5 km. To look at how this result is sensitive to the threshold reflectivity, we reprocessed the results with the 0 dBZ threshold. By lowering the threshold to 0 dBZ, an increment of ~1 km in the vertical extension of CFAD is found in the model, but the echo top height of the observation is not changed much. As a result, the choice of threshold does not change the conclusion of severe model underestimation in echo top height."

But what do you learn with this about the deficiencies in the model, especially about the convective vertical velocities and convective mass flux assumptions. You could check, maybe. There is a very good discussion on this in the Labbouz et al. (2018) paper. This paper also tackles similar issues but using different types of comparisons, so I believe it should be quoted in your paper (as well as references therein).

The literature recommended and references therein provided an in-depth discussion of how to improve the modeling of convective clouds in GCMs to better match radar retrievals. In this study, we are not aiming for that purpose. We conducted the direct comparison of reflectivity between model and radar to identify model biases, and we did some tests by tuning a series of parameters in the ZM cumulus scheme and cloud microphysics scheme to see if the large biases in the echo top height can be alleviated. The results of this study can provide metrics for evaluating the cumulus parameterizations or provide insights for improving the cumulus parameterizations, which would be nice follow-on work.

We have cited the paper and provided a discussion about the further work at the end of Section 4. "In addition, the results of this study can provide metrics for evaluating the cumulus parameterizations or provide insights for further improving the cumulus parameterizations like Labbouz et al. (2018), which can be a follow-on work."

On another hand if you take the model cloud top height, chances are that the NEXRAD radars won't have the sensitivity to detect it, and this time the radar statistics will show lower cloud top heights than the model (I think I have seen statistics somewhere showing a systematic ~ 2km difference between cloud top height and 0-dBZ echo top heights, but I can't find the reference just yet). If it does not though, it would mean that the model really underestimates cloud top heights, which would be very interesting.

We would like to clarify that we only focused on the echo top height and did not look at cloud top height. Evaluating echo top height allows us to know that the model failed to simulate the occurrence of large ice-phase particles at high levels in deep convective clouds. As the reviewer mentioned, NEXRAD won't have enough sensitivity to detect the cloud top height so we cannot compare the observed echo top height with the modeled cloud top height. To ensure a fair comparison, the same radar threshold has to be applied. We tested the sensitivity of our results with a different threshold (0 dBZ) as shown in Fig. R1, and the model still severely underestimates the echo top height.

**Specific comments:**

1. The 13.6 GHz versus 3 GHz difference: my only issue with this is that you should make sure that you have switched off any Ku-band attenuation correction in COSP. Have you? If so you should mention it in the paper.

We actually turned on the Ku-band attenuation correction in COSP, and we believe this is still a valid comparison. First, Rayleigh scattering is satisfied at 13.6 GHz frequency with respect to gases and most ice/liquid particles, thus the attenuation correction making no differences for those hydrometeor species. Secondly, it is extremely difficult for the global model to simulate ice particles with a size large enough to be comparable with the wavelength (~2 cm), which has been discussed in the CFAD comparison. Last and the most important, the COSP mimics the satellite view from space to the ground, therefore the layer most vulnerable to the attenuation caused by large precipitation droplets is close to the ground (i.e., 1 km), which has been excluded from the comparison. The clarification has been added, "In the COSP simulator, the 13.6 GHz frequency ensures the Rayleigh scattering calculation. Although an attenuation correction has been applied, because the COSP mimics the satellite view from space to the ground, the layer below 1-km altitude is most vulnerable to attenuation caused by large precipitation particles, which has been excluded from the comparison."

2. Line 139: it does not make sense to say that the minimum detectable reflectivity of NEXRAD radars is 0 dBZ. The MDR varies with range (MDR(range)=MDR(1km) + 20 log10(range_km)), so you either need to provide the sensitivity at 1 or 10 or 100km (whatever you prefer) or state that a hard threshold of 0 dBZ is applied somewhere in the NEXRAD processing. Also, I would be surprised if the NEXRAD radars can detect 0 dBZ at 250km range (but I don't have those numbers).

We didn't directly use the original NEXRAD scan data, but the gridded 3D mosaic data. The 0 dBZ should not be the threshold of the NEXRAD but the threshold of the dataset we use. A correction has been made in the text, "as shown in previous studies (e.g., Wang et al., 2015, 2016, 2018; Feng et al., 2012, 2019), the minimum reflectivity of the 3D mosaic NEXRAD dataset is 0 dBZ (Fig. 1a)."

3. Back to the 8-dBZ threshold: how sensitive are all your results to the use of 8-dB threshold? What happens if you take 0 dB (you say NEXRAD can detect 0dB), so why didn't you use 0 dB to get closer to the true cloud top? It would make more sense in my opinion.

**See our response to your major comment above.**

4. Line 180-181: again, you say here that your most distinct result is about echo top height. But I believe it does not teach you anything about potential model deficiencies, including the convective vertical velocities and mass flux issues highlighted in Labbouz et al. (2018) and others. Your main result is actually that the model is underestimating reflectivities, but may well be excellent at producing realistic cloud tops. The only thing is that they have lower reflectivities than in reality.

**See our response to the major comment above.**

5. Line 194: this statement is likely wrong. I would bet that it simulates reflectivities at heights greater than 11km, but they are under your 8 dB threshold due to ice microphysics deficiencies.

**We used a lower threshold 0 dBZ and the conclusion is not affected, as seen from our response to your major comment.**

6. Line 226: ~5dBZ: it is more like 4 dB and only for Z<25 dBZ, so maybe this statement should be modified to reflect this.

**We have modified the statement to "In addition, the modified microphysics assumptions produce higher values of reflectivity, in better agreement with observations, and the grid-mean radar reflectivities increase by ~4 dBZ (Fig. 3) mainly for values less than 25 dBZ.**

7. Line 323: typo "Brodzik"

**The correction has been made.**

8. Figure 2 and associated discussion in section 3: comparing mean reflectivities is interesting, but only one aspect of what you'd like to get right with a model. The two main other things I would personally try to assess is the standard deviation of reflectivity at 100km scale (to check if the model reproduces the observed variability even if it does not have the mean right) and the 95th percentile or even 99th percentile if you have enough samples (to check if the model has any skills in forecasting extremes). These two additional things would greatly enhance the scientific content and scope of this model evaluation exercise.

**We have added a Table to the manuscript (Table 2) to include the standard deviation and the 95th percentile values. The discussion has been added correspondingly in Section 3.2 and the following sentence has been added to the Conclusion. "EAMv1 can simulate the variability and extreme value of reflectivity at the lower troposphere but significantly underestimate them at high levels."**

References:

*Labbouz, L., Z. Kipling, P. Stier, and A. Protat, 2018: How well can we represent the spectrum of convective clouds in a climate model? Journal of the Atmospheric Sciences, 75(5), 1509–1524.*

Good luck with the review,

Alain Protat

Melbourne, 25/06/2020

---

## Referee Report (RR1)

**Review of "Using Radar Observations to Evaluate 3D Radar Echo Structure Simulated by the Global Model E3SM Version 1" by Jingyu Wang et al.**

I've been drafted in and did not review the original version of this paper, but I agree with the reviewers' original compliments regarding its readability and usefulness. I judge that the authors address the original reviews and the addition of the 0 dBZ threshold sensitivity test is particularly helpful for interpretation.

The authors reject reviewer 1's suggestion to analyse uncoupled runs and I side fully with the authors' response. The submitted paper explores issues with physical parameterisations and COSP within the model, this alone is a large undertaking that would be hindered by additional model-observation discrepancies introduced via natural variability in free-running simulations. The new version of the text makes this argument succinctly.

I have one technical comment regarding Figure 2 that I believe must be addressed for accuracy, and request that the authors cover some additional ground in the discussion to really tie this paper up. I also put a set of "minor" and "very minor" points where I suggest grammar or text to reduce ambiguity.

I am requesting only minor changes and I don't think the conclusions will be strongly affected. I think the authors have done a nice job with this paper and would support publication after these changes.

**Main points**
   1. **Figure 2**

My understanding is that Figure 2 shows "sub-grid" statistics, i.e. there are $N$ grid cells included, and the observational histograms contain $625N$ entries while the model histograms contain $50N$ entries. Isn't this like comparing the histograms of properties at different spatial resolutions? The histogram from the smaller-grid-cell sample (i.e. observations) would generally be broader anyway, so the two cannot be directly compared. Instead, the observations should be averaged somehow in order to provide 50 per grid cell.

If I am wrong about this then please clarify in the text.

Fortunately, it doesn't look like your overall conclusions would be strongly affected.

   2. **Discussion**

Some potential limitations of the analysis are not covered, primarily related to the model-observation comparison.

I haven't used COSP for surface measurements but I presume your assumed viewing geometry is for an upward-pointing radar. Please provide the assumed viewing geometry in Section 2.2 and in the discussion you should cover whether this is somehow addressed or if it may introduce model-observation discrepancies.

For example, are there regions in which topography may affect the NEXTRAD data like it can do for precipitation frequency (e.g. Smalley et al., 2017: https://doi.org/10.1175/JHM-D-16-0242.1)? Probably not for your big central MCS regions, but maybe in others.

Is attenuation a problem? If your simulations are looking up and you compare with a NEXRAD radar that's looking side on to a convective storm, then perhaps model pulse gets attenuated by the dense convective core so you see lower high-altitude dBZ than you would get from the NEXRAD radar. It's my understanding that even S-band upper-level dBZ can be affected in extreme cases (e.g. lots of big particles in the melting layer), and these extreme cases might contribute a substantial fraction of the very high altitude/high dBZ results.

Finally, the discussion might also benefit from laying out the primary other factors which could contribute to your model-observation differences. For example; you modified the COSP particle size distribution and see changes, but who's to say it's now realistic? What features of the model-observation discrepancy could perhaps be explained by these factors; how far can you exclude them; and are there obvious tests for future studies to address and rule out such factors? I remain concerned about the 13.6 GHz to 3 GHz change too, and this should be mentioned again in the Discussion section to remind readers. Putting these things all together will make it easier for the community to contextualise and use your results.

I insist on commentary about the viewing geometry and its expected implications for the results, the rest of the potential expanded discussion I leave to the authors' discretion.

**Minor points**
**P2L54—56:**
"As discussed by Iguchi et al. (2018), precipitating ice particles have a large variation in habits and scattering properties, and the effect of non-Rayleigh scattering and multiple scattering by large precipitating ice particles could introduce large uncertainty into simulating the cold-season radar reflectivity field. To avoid this uncertainty, we examine only the warm season of the three years from 2014 to 2016."

I interpret that as saying that it is only in the cold season that you see (i) ice particle scattering, (ii) non-Rayleigh scattering and (iii) multiple scattering. Specifically, you "avoid" it (i.e. it is zero) during the warm season. I think it happens sometimes in warm season storms. Suggestion:

"As discussed by Iguchi et al. (2018), precipitating ice particles have a large variation in habits and scattering properties, and the effect of non-Rayleigh scattering and multiple scattering by large precipitating ice particles could introduce large uncertainty into simulating the radar reflectivity field. To reduce uncertainty due to these factors, we examine only the warm season of the three years from 2014 to 2016."

**P4L108:**
"…The detailed documentation of those changes is in Table 1…" this table is appreciated and efficiently carries important information. I would like to see all the Default values, where there are no changes you could insert "-" in the Modified columns. This would provide complete information and visually guide the reader to identify where changes have been made.

The top two rows report Gamma distributions with "width" of "0". What is this width? Since you're already reporting the mean then I think it would be consistent to insert the variance in the final column and then add text to the caption to explain this is what you've done.

If I have misunderstood, then please expand the caption to avoids such misunderstandings.

**P5L130/131 and PL140:**
"…we nevertheless perform the Gaussian smoothing of GridRad data to match the model time step (30 min) in the comparison." AND "The simulation data are saved hourly, consistent with the hourly GridRad data."

I can't understand this – these sentences appear contradictory. Please clarify. For interpreting the results I assumed the second description applied.

**P5L136:**
"We also did the test with 0 dBZ to look at the sensitivity of our key results to the choice of the threshold value. Thus, after coarsening the 4-km GridRad data to a model grid element, only the grid elements with a mean value larger than 8 dBZ are taken into account in both observations (Fig. 1b) and simulation (Fig. 1c)."

This is the first time you mention the sensitivity test and I think it could be clearer. Example suggestion:

"We also tested with a threshold of 0 dBZ and report later on how it only has minor effects on our conclusions. For our main results, after coarsening the 4-km GridRad data to a model grid element, only the grid elements with a mean value larger than 8 dBZ are taken into account in both observations (Fig. 1b) and in the simulation (Fig. 1c)."

**Very minor points**
P1L30: "Over the continental U.S." would be a good point to introduce the "CONUS" acronym which is used later without expansion.

P2L31/32: please insert wavelength or frequency after S-band here. The earlier the better.

P2L49: "over the CONUS for the three years (2014-2016)" parentheses are jarring, please remove.

P2L52: "Over the CONUS, warm-season is dominated by convective processes". The hyphen makes me think "warm-season" is intended as a compound modifier so I guess you're just missing the word "precipitation".

P3L76—77: "pressure-based terrain following coordinate" – you could optionally also insert "hybrid sigma" descriptor here to introduce it earlier than Section 2.

P3L89: "…spaceborne satellites…" typo: "satellites" is an adjective here so should be singular.

P3L92: "…direct measurements form 3D scanning radars…" typo: I think you mean "…from 3D…"

P4L95: "…pseudoobservations using forward calculation" typo: missing article or pluralisation (e.g. "forward calculations").

P5L130: "we nevertheless perform the Gaussian smoothing" typo: I don't think you need "the" before "Gaussian smoothing".

P6L189: "Meanwhile, the modeled standard deviation and the extreme values are smaller, indicating the model has a difficulty to capture the observed verifiability." – I don't understand "observed verifiability" and I'd replace "to capture" with "capturing".

P7L208/209: "For the reflectivity >35 dBZ, simulation has a higher probability". Looks like "the" is in the wrong place, I think it should be: "For reflectivity >35 dBZ, the simulation has a higher probability…"

P7L211: "the percentile values are consistent between model and observations". Looks like missing "the" before "model". There are some other cases like this, just keep an eye out for that when you skim through.

P7L222: "lowering the threshold to 0 dBZ, an increment of ~1 km in the vertical extension of CFAD is found in the model, but the echo top height of the observation". I think "the" is needed before CFAD (it is not a proper noun, there are many CFADs) and "observation" should be pluralised here.

P8L246—248 "Xie et al. (2019) improved the diurnal cycle of precipitation in E3SM v1 recently by modifying the convective trigger function in the ZM scheme. It will be interesting to see if it can simulate the double-peaks in observed column-maximum reflectivity in the future.". This is interesting and useful context, good inclusion.

P8L249: "3.4 Sensitivity of Simulated Echo Top Height Tunable Parameters of the Global Model" I had trouble parsing this. Do you mean sensitivity of simulated echo top hight *to* tunable parameters?

P8L250: "Different from the model evaluation of". This seems grammatically weird to me, perhaps "Differently from…", although I'd probably pick "Compared with…"

P8L254—255: "tunable parameters as listed in Table 3. Each test is based on the default setup for all other parameters.". This sentence makes sense but thanks to the structure I re-read it a couple of times to be sure I'd understood it. I suggest something more explicit, like "In each test a single parameter is changed, and all other parameters retain their default values".

P9L276—277: "In summary, changing any single parameter alone in the ZM scheme does not improve the simulation of echo top height." Did you change all the parameters, or is this a select subset? If a select subset then I think you should specify here: "changing any of our selected parameters individually in the…"

P9L278—279: "(i.e., those resolved by model resolution)." Is repetitive, how about "those resolved by the model"?

P9L280: "and precipitation by changing the large-scale forcing on which cumulus clouds are calculated" it sounds unnatural to me that clouds are calculated "on" a forcing. How about: "…the large-scale forcing which feeds into the cumulus cloud calculations".

P9L283: "Attempts of accelerating" I think should be "attempts at accelerating".

P9L287: "only gains 500-800 m increment" missing "a" before "500—800 m"

P10L301: "With default microphysics assumptions" I think this would make more sense as "the default microphysics assumptions", since you're referring to the individual set of assumptions in this model, rather than assumptions in general.

P11L1—2: "circulation is nudged towards observations for the simulations in this study, which represents the upper bound of model performance." Again the phrasing is ambiguous here to me, because it's not clear what the "which" refers to among all the nouns in the earlier part of the sentence (nudging? Circulation? Simulations?). My first choice would be to remove everything after the comma because the next two sentences explain it, but if you really want to keep that bit then how about "…for the simulations in this study, so our results represent the best-case model performance".

---

## Author Response (AR2)

Dear editor and referees:

Thank you very much for the advice to improve the quality of this manuscript. We carefully addressed all the comments and made corresponding changes to the manuscript. In this "response to comments" document, we provided detailed responses in blue bold as below.

**Referee #1: Peter May, peter.may@monash.edu**

I am broadly happy with the paper and responses, except for the following.

1. Line 48: "Our goal is to provide a comprehensive evaluation of both horizontal pattern and vertical structure of cloud and precipitation." – How can you do this if you do not discriminate between convective, stratiform and large scale stratiform processes given the fundamentally different processes, heating and drying profiles. Radar simulations on such large grid sizes are already problematic, but given a key part of a convective paramaterisation is the fraction of the grid, surely the model and interpretation should could be improved. This is clearly beyond the current work, but should be acknowledged and planned for future work.

**We have added the discussion about future work. "Future studies can also focus on separately evaluating properties in convective and stratiform regions, since the thermodynamic and reflectivity profiles are fundamentally different between the two regions."**

2. CFADS without considering this are limited in value in my opinion. Processing the radar data for convective and stratiform fractions in observations is straightforward (e.g. Steiner's algorithm) and readily compared with convective fraction in cumulus paramaterisations and would provide a clear test for the models.

**Convective cloud fraction is not parameterized in mass flux-based convection schemes including the ZM scheme. It is assumed to be <<1 for typical GCM resolutions such as at 1-degree grid spacing or coarser. Since radiative transfer calculations need it, convective cloud fraction is separately diagnosed in these schemes. In the ZM scheme, it is fitted to be a function of cloud mass flux, thus should be viewed as a tuning parameter for cloud radiation calculation. As such, it is not very meaningful scientifically to evaluate it in the current ZM scheme. However, in the future if it becomes an independent variable in a convection scheme (for instance, for grey-zone resolutions convective cloud fraction will be needed in parameterizations), then evaluating it will be meaningful.**

3. L154. "In EAMv1, 50 sub-columns are used for calculating the mean radar reflectivity for a model grid box. There are 625 pixels inside each 1° grid for NEXRAD data to provide a probability density function (PDF) of observed reflectivity within the box. F. " – how can this be reliably done if you do not know what fraction is convective given radically different reflectivity profiles and magnitudes – even for similar rain rates?

**We didn't consider the convective fraction when calculate the grid-mean reflectivity. All the subgrid reflectivity values and NEXRAD pixels within each 1° grid box are linearly average with no discrimination. We understand the reflectivity profiles are significantly different between stratiform and convective, but the COSP calculates the subgrid reflectivity independent of what E3SM uses. More importantly, the convective cloud fraction is not parameterized in mass flux-**

**based convection schemes including the ZM scheme (assumed to be <<1 for typical GCM resolutions such as at 1-degree grid spacing or coarser), therefore its evaluation is not very meaningful. The clarification has been added, "in addition, the subgrid distribution results from COSP are calculated based on the assumption about the distribution of cloud and precipitation among the 50 subcolumns, which is independent of what E3SM uses. Therefore, a higher-order consistency between the COSP and the host model is warranted in future studies. In this following analysis, we focus on the evaluation of the simulated 3D radar reflectivity field at the model's native grid, which is 1-degree, since the subgrid information from COSP does not directly reflect how E3SM does it. Also, the convective cloud fraction is not parameterized in mass flux-based ZM scheme and is diagnosed from cloud mass flux for cloud radiation calculation, which is treated as a tunable parameter, whose evaluation is not very meaningful unless it becomes an independent variable, for instance, for grey-zone resolutions."**

4. L211 –" the reflectivity below 4 km is consistent", but in the CFAD's the model has an odd double peak in the reflectivity and low numbers of low reflectivity compared with the observations. Some of this could be associated with the downscaling, but could also be due to the lack of proper classifications when downscaling the model reflectivity. I think these differences are significant and potentially important. I would also note identifying issues such as this is exactly the point of papers such as this.

**We have discussed the discrepancy of CFADs' overall shape between the model and observations and related this to the lack of separation between convective and stratiform. However, we think averaging to 1° should have more impact on the observation than the model as mentioned by Referee 4's minor comment 7 since the double peak shown in the model is only related to the choice of display interval. As a result, the following discussion has been added, "Regarding the overall shape of CFADs, the model follows the well-known pattern where the reflectivity value range of high frequency zone (> 3.2%) increases from cloud top to the freezing level, and then slowly decreases or remains constant below the freezing level. The cores of maximum frequency (> 5%) are located in the centres of the high frequency zones. However, these characteristics are not presented in the observations, whose high frequency zones are greatly skewed to the lower reflectivity values. The characteristics of NEXRAD's CFADs could be due to averaging from fine resolution (4 km) to coarse resolution (1°), as well as averaging of convective and stratiform components because the two components produce significantly different reflectivity profiles and magnitudes."**

5. Figure 6 showing the diurnal cycle looks very odd. In the model there is essentially none despite the well known biases in global models for initial deep convection too early in the day and the observations show two very narrow peaks that certainly are not consistent with previous observations such as the cited Carbone and Tuttle paper. As the authors say, it may represent issues in triggering convection in the model, but without information on convective fraction it is difficult to know. Note that Carbone and Tuttle showed that the timing of diurnal maxima was longitude dependent with propagating modes that would smear the diurnal cycle. However, what you have plotted, the reflectivity maxima in the model may not be expected to vary much as long as there is some precipitation given the profile is parameterised from the existence of rainfall. Frankly, I do not know what to make of the peaks in reflectivity in the observations. These seem very narrow. On what spatial scale are these reflectivity maxima and how has this been averaged

in time? If it is on the 4 km grid they may be too low, depending on how the temporal sampling is done. For the moment, I am not sure what this figure adds and would delete it but keep some of the text noting the diurnal variations (that are discussed in your previous paper? ).

**The figure of diurnal cycle has been removed, but we keep some discussion as suggested. "As evaluated in Zheng et al. (2019), E3SM v1 failed to simulate the diurnal variation of precipitation over the central United States, where the observed nocturnal peak is greatly underestimated. Xie et al. (2019) improved the diurnal cycle of convection in E3SM v1 recently by modifying convective trigger function in the ZM scheme. It will be interesting to see if the 3D radar reflectivity fields can be better simulated using the updated ZM scheme".**

**Referee #3: Anonymous**

I've been drafted in and did not review the original version of this paper, but I agree with the reviewers' original compliments regarding its readability and usefulness. I judge that the authors address the original reviews and the addition of the 0 dBZ threshold sensitivity test is particularly helpful for interpretation.

The authors reject reviewer 1's suggestion to analyse uncoupled runs and I side fully with the authors' response. The submitted paper explores issues with physical parameterisations and COSP within the model, this alone is a large undertaking that would be hindered by additional model-observation discrepancies introduced via natural variability in free-running simulations. The new version of the text makes this argument succinctly.

I have one technical comment regarding Figure 2 that I believe must be addressed for accuracy, and request that the authors cover some additional ground in the discussion to really tie this paper up. I also put a set of "minor" and "very minor" points where I suggest grammar or text to reduce ambiguity.

I am requesting only minor changes and I don't think the conclusions will be strongly affected. I think the authors have done a nice job with this paper and would support publication after these changes.

**We thank the referee for providing detailed line-by-line comments, which greatly helped to improve the readability of the manuscript.**

**Main points**

**1. Figure 2**

My understanding is that Figure 2 shows "sub-grid" statistics, i.e. there are N grid cells included, and the observational histograms contain 625N entries while the model histograms contain 50N entries. Isn't this like comparing the histograms of properties at different spatial resolutions? The histogram from the smaller-grid-cell sample (i.e. observerations) would generally be broader anyway, so the two cannot be directly compared. Instead, the observations should be averaged somehow in order to provide 50 per grid cell.
If I am wrong about this then please clarify in the text.
Fortunately, it doesn't look like your overall conclusions would be strongly affected.

**All NEXRAD data are averaged from 625 samples to 50 samples to match the simulation, then the PDF are generated accordingly. Figure 3 is also updated to account for the problems noted by Referee 4's minor comment 5. We have added "After averaging the NEXRAD pixels at subgrid scale to 50 samples to match the COSP's subcolumns, Fig. 3 compares the simulated subgrid reflectivity distribution to the NEXRAD distribution based on all the GridRad samples combined for the 3-year period at each individual level, where the interval of reflectivity bins is 1 dBZ."**

**2. Discussion**

Some potential limitations of the analysis are not covered, primarily related to the model-observation comparison. I haven't used COSP for surface measurements but I presume your assumed viewing geometry is for an upwardpointing radar. Please provide the assumed viewing geometry in Section 2.2 and in the discussion you should cover whether this is somehow addressed or if it may introduce model-observation discrepancies.

For example, are there regions in which topography may affect the NEXTRAD data like it can do for precipitation frequency (e.g. Smalley et al., 2017: https://doi.org/10.1175/JHM-D-16-0242.1)? Probably not for your big central MCS regions, but maybe in others.
**The viewing geometry used in the COSP is not like an upwardpointing radar. As stated in the Section 2.3, the viewing geometry of the COSP mimics that of satellite, which is from space to the ground. Clarifications has been added to "the COSP mimics the satellite view from space to the ground, thus the impact of topography is not an issue as ground-based radars (Smalley et al., 2017). With the downward viewing geometry, the layer below 1-km altitude is most vulnerable to the possible attenuation caused by large precipitation particles, which has been excluded from the comparison."**

Is attenuation a problem? If your simulations are looking up and you compare with a NEXRAD radar that's looking side on to a convective storm, then perhaps model pulse gets attenuated by the dense convective core so you see lower high-altitude dBZ than you would get from the NEXRAD radar. It's my understanding that even S-band upper-level dBZ can be affected in extreme cases (e.g. lots of big particles in the melting layer), and these extreme cases might contribute a substantial fraction of the very high altitude/high dBZ results.
**As clarified in the comment above, the viewing geometry of the COSP is from space to the ground. This said, the attenuation caused by near-surface convective core has no impact on the high-altitude reflectivities. Moreover, we stated in the text "the layer below 1-km altitude is most vulnerable to the possible attenuation caused by large precipitation particles, which has been excluded from the comparison."**

Finally, the discussion might also benefit from laying out the primary other factors which could contribute to your model-observation differences. For example; you modified the COSP particle size distribution and see changes, but who's to say it's now realistic? What features of the model-observation discrepancy could perhaps be explained by these factors; how far can you exclude them; and are there obvious tests for future studies to address and rule out such factors? I remain concerned about the 13.6 GHz to 3 GHz change too, and this should be mentioned again in the Discussion section to remind readers. Putting these things all together will make it easier for the community to contextualise and use your results.
**The reviewer might be misunderstanding what we did here. What we did was to make the microphysics assumptions used in the COSP to be consistent with those in the cloud microphysics scheme of the host model, which was just about fixing a problem. Fixing the problem indeed helped but the model is still significantly biased in the subgrid distribution, which might contribute by other issues. We have added discussion related to this. "Although the simulated subgrid reflectivity distribution is improved by setting the microphysics assumptions used in COSP consistent with the MG2, the model is still significantly biased. In addition to the intrinsic model-observation differences in the number concentrations and mixing ratios of hydrometeors, there are other possible error sources related to the reflectivity calculation as mentioned in Section 2.2. For example, (1) the mixing ratios are not directly passed from the host model to COSP, instead are converted from the model's precipitation fluxes, (2) the spectral parameters for defining a Gamma**

**distribution are not consistent from MG2, and (3) the assumptions of subgrid distribution and hydrometeor vertical overlap are simple and not consistent with other parts of the host model. In addition, the subgrid distribution results from COSP are calculated based on the assumption about the distribution of cloud and precipitation among the 50 subcolumns, which is independent of what E3SM uses. Therefore, a higher-order consistency between the COSP and the host model is warranted in future studies."**

**For the concern of the 13.6 GHz vs. 3 GHz, we performed a series of offline tests of COSP simulation using the frequency of 3 GHz, 13.6 GHz, and 94 GHz. The comparisons of their corresponding reflectivities are shown in Fig. 1. As shown, the reflectivities values with 3 GHz are very similar to those with 13.6 GHz, indicating the Rayleigh scattering is satisfied for both frequencies in this application. Note the particle size simulated by global model at 1-deg scale is known much smaller than the reality, with diameters far smaller than 2.2 cm which leads to the similar reflectivity simulation for any frequency lower than 13.6 GHz.**

I insist on commentary about the viewing geometry and its expected implications for the results, the rest of the potential expanded discussion I leave to the authors' discretion.
**The discussion of viewing geometry has been added to the Section 2.3.**

**Minor points**

**P2L54—56:**
"As discussed by Iguchi et al. (2018), precipitating ice particles have a large variation in habits and scattering properties, and the effect of non-Rayleigh scattering and multiple scattering by large precipitating ice particles could introduce large uncertainty into simulating the cold-season radar reflectivity field. To avoid this uncertainty, we examine only the warm season of the three years from 2014 to 2016."
I interpret that as saying that it is only in the cold season that you see (i) ice particle scattering, (ii) non-Rayleigh scattering and (iii) multiple scattering. Specifically, you "avoid" it (i.e. it is zero) during the warm season. I think it happens sometimes in warm season storms.
Suggestion:
"As discussed by Iguchi et al. (2018), precipitating ice particles have a large variation in habits and scattering properties, and the effect of non-Rayleigh scattering and multiple scattering by large precipitating ice particles could introduce large uncertainty into simulating the radar reflectivity field. To reduce uncertainty due to these factors, we examine only the warm season of the three years from 2014 to 2016."
**We agree that large ice particles could definitely occur in the warm season. The text has been modified as suggested.**

**P4L108:**
"…The detailed documentation of those changes is in Table 1…" this table is appreciated and efficiently carries important information. I would like to see all the Default values, where there are no changes you could insert "-" in the Modified columns. This would provide complete information and visually guide the reader to identify where changes have been made.
**All the default values are added, and the table is modified as suggested.**

The top two rows report Gamma distributions with "width" of "0". What is this width? Since you're already reporting the mean then I think it would be consistent to insert the variance in the final column and then add text to the caption to explain this is what you've done.

If I have misunderstood, then please expand the caption to avoids such misunderstandings.

**The width means the shape parameter in Gamma distribution for describing the dispersion of the distribution. This is clarified as a footnote of the table. A fixed value is used in two-moment microphysics schemes, so here we made it to be consistent with MG2.**

**P5L130/131 and PL140:**

"…we nevertheless perform the Gaussian smoothing of GridRad data to match the model time step (30 min) in the comparison." AND "The simulation data are saved hourly, consistent with the hourly GridRad data."

I can't understand this – these sentences appear contradictory. Please clarify. For interpreting the results I assumed the second description applied.

**The model output is at hourly frequency, but the model time step is 30 min. Therefore the hourly radar reflectivity field represent the average state of the past 30 min, based on which the GridRad data are smoothed to gap the model-observation temporal mismatch.**

**P5L136:**

"We also did the test with 0 dBZ to look at the sensitivity of our key results to the choice of the threshold value. Thus, after coarsening the 4-km GridRad data to a model grid element, only the grid elements with a mean value larger than 8 dBZ are taken into account in both observations (Fig. 1b) and simulation (Fig. 1c)."

This is the first time you mention the sensitivity test and I think it could be clearer. Example suggestion:

"We also tested with a threshold of 0 dBZ and report later on how it only has minor effects on our conclusions. For our main results, after coarsening the 4-km GridRad data to a model grid element, only the grid elements with a mean value larger than 8 dBZ are taken into account in both observations (Fig. 1b) and in the simulation (Fig. 1c)."

**Modification has been made as suggested.**

**Very minor points**

P1L30: "Over the continental U.S." would be a good point to introduce the "CONUS" acronym which is used later without expansion.

**Modification has been made accordingly.**

P2L31/32: please insert wavelength or frequency after S-band here. The earlier the better.

**Modification has been made accordingly.**

P2L49: "over the CONUS for the three years (2014-2016)" parentheses are jarring, please remove.

**The parentheses has been removed as suggested.**

P2L52: "Over the CONUS, warm-season is dominated by convective processes". The hyphen makes me think
"warm-season" is intended as a compound modifier so I guess you're just missing the word "precipitation".
**Modifications has been made as suggested.**

P3L76—77: "pressure-based terrain following coordinate" – you could optionally also insert "hybrid sigma" descriptor here to introduce it earlier than Section 2.
**Modification has been made accordingly.**

P3L89: "…spaceborne satellites…" typo: "satellites" is an adjective here so should be singular.
**Correction has been made.**

P3L92: "…direct measurements form 3D scanning radars…" typo: I think you mean "…from 3D…"
**Correction has been made.**

P4L95: "…pseudoobservations using forward calculation" typo: missing article or pluralisation (e.g. "forward calculations").
**Correction has been made.**

P5L130: "we nevertheless perform the Gaussian smoothing" typo: I don't think you need "the" before "Gaussian smoothing".
**Correction has been made.**

P6L189: "Meanwhile, the modeled standard deviation and the extreme values are smaller, indicating the model has a difficulty to capture the observed verifiability." – I don't understand "observed verifiability" and I'd replace "to capture" with "capturing".
**It is "variability" that we intended to use. The typo has been corrected.**

P7L208/209: "For the reflectivity >35 dBZ, simulation has a higher probability". Looks like "the" is in the wrong place, I think it should be: "For reflectivity >35 dBZ, the simulation has a higher probability…"
**Modifications has been made as suggested.**

P7L211: "the percentile values are consistent between model and observations". Looks like missing "the" before "model". There are some other cases like this, just keep an eye out for that when you skim through.
**The missing "the" is added. Similar corrections have been made throughout the entire manuscript with careful examination.**

P7L222: "lowering the threshold to 0 dBZ, an increment of ~1 km in the vertical extension of CFAD is found in the model, but the echo top height of the observation". I think "the" is needed before CFAD (it is not a proper noun, there are many CFADs) and "observation" should be pluralised here.
**Modifications has been made as suggested.**

P8L246—248 "Xie et al. (2019) improved the diurnal cycle of precipitation in E3SM v1 recently by modifying the convective trigger function in the ZM scheme. It will be interesting to see if it can simulate the double-peaks in observed column-maximum reflectivity in the future.". This is interesting and useful context, good inclusion.
**Thank you.**

P8L249: "3.4 Sensitivity of Simulated Echo Top Height Tunable Parameters of the Global Model" I had trouble parsing this. Do you mean sensitivity of simulated echo top hight *to* tunable parameters?
**Yes, a "to" has been added.**

P8L250: "Different from the model evaluation of". This seems grammatically weird to me, perhaps "Differently from…", although I'd probably pick "Compared with…"
**We have changed "different" to "differently", because the two types of evaluation are truly different.**

P8L254—255: "tunable parameters as listed in Table 3. Each test is based on the default setup for all other parameters.". This sentence makes sense but thanks to the structure I re-read it a couple of times to be sure I'd understood it. I suggest something more explicit, like "In each test a single parameter is changed, and all other parameters retain their default values".
**Thank you for the suggestion, modifications have been made accordingly.**

P9L276—277: "In summary, changing any single parameter alone in the ZM scheme does not improve the simulation of echo top height." Did you change all the parameters, or is this a select subset? If a select subset then I think you should specify here: "changing any of our selected parameters individually in the…"
**It is a selected subset. Modifications have been made accordingly.**

P9L278—279: "(i.e., those resolved by model resolution)." Is repetitive, how about "those resolved by the model"?
**Modifications have been made accordingly.**

P9L280: "and precipitation by changing the large-scale forcing on which cumulus clouds are calculated" it sounds unnatural to me that clouds are calculated "on" a forcing. How about: "…the large-scale forcing which feeds into the cumulus cloud calculations".
**Modifications have been made accordingly.**

P9L283: "Attempts of accelerating" I think should be "attempts at accelerating".
**Modification has been made accordingly.**

P9L287: "only gains 500-800 m increment" missing "a" before "500—800 m"
**Modification has been made accordingly.**

P10L301: "With default microphysics assumptions" I think this would make more sense as "the default microphysics assumptions", since you're referring to the individual set of assumptions in this model, rather than assumptions in general.
**"The" has been added as suggested.**

P11L1—2: "circulation is nudged towards observations for the simulations in this study, which represents the upper bound of model performance." Again the phrasing is ambiguous here to me, because it's not clear what the "which" refers to among all the nouns in the earlier part of the sentence (nudging? Circulation? Simulations?). My first choice would be to remove everything after the comma because the next two sentences explain it, but if you really want to keep that bit then how about "…for the simulations in this study, so our results represent the best-case model performance".

**Good suggestion. Modifications have been made accordingly.**

**Referee #4: Anonymous**

Summary of paper:
The manuscript presents results of an evaluation of the E3SM model against NEXRAD radar observations for the summer periods during 2014-2016. The authors used the COSP forward simulator package to generate radar reflectivity values from their model cloud fields and averaged both the sub-grid COSP output and the NEXRAD observations to a 1-degree horizontal grid and 1-km vertical grid for like-with-like comparison. The model average reflectivity exceeds the observed value slightly at 2-km height, but at heights above 4 km the model generally does not produce enough cloud above the threshold reflectivity value. Sensitivity testing considering the convection and cumulus parameterisations does not improve this model bias.

Review summary:
This is generally a well-written paper with good quality figures. The evaluation of NWP models against 3D cloud and precipitation observations is of great importance and the present evaluation against NEXRAD is novel. The methodology is incomplete or insufficiently justified in places, which leads to serious concerns about the results. Nevertheless, these concerns might be overcome with appropriate clarifications or revisions and as such publication may be considered after major corrections.
**We thank the referee for the accurate summary of our study and the detailed suggestions and comments.**

Major comment 1:

The study is hindered by its original objective to evaluated the model against GPM and hence the implementation of the 13.6 GHz frequency in COSP. There are two issues at stake, namely (a) whether the comparison of 13.6 GHz simulated reflectivity against S-band (3-GHz) is appropriate and (b) whether the implementation has been done appropriately.

(a) The authors justify the 13.6 GHz versus 3 GHz comparison by citing their Wang et al. (2019b) study. While that is a nice paper, it is not a sufficiently comprehensive evaluation of the 13.6 GHz reflectivity against the 3 GHz reflectivity to convince the reader that these two are interchangeable. The key figure in that paper (Figure 2) uses normalisation, which removes any excess (or deficit) in cloud detection, which is of importance for this study. The normalisation within cloud, also performed in that Figure, masks the reduction in reflectivity values obtained with GPM (13.6 GHz) both due to attenuation and due to Mie scattering.

Beyond this general unease with the comparison, there are various studies that suggest a necessary conversion from Ku (13.6 GHz) to S band, with different equations used for ice and liquid phases. In particular, recent studies using the GPM radar to calibrate ground-based radars use such conversions:

Warren, R. A., A. Protat, S. T. Siems, H. A. Ramsay, V. Louf, M. J. Manton, and T. A. Kane, 2018: Calibrating Ground-Based Radars against TRMM and GPM. J. Atmos. Oceanic Technol., 35, 323–346, https://doi.org/10.1175/JTECH-D-17-0128.1.

To answer the first question "whether the comparison of 13.6 GHz simulated reflectivity against S-band (3-GHz) is appropriate?", we performed a series of offline tests of COSP simulation using the frequency of 3 GHz, 13.6 GHz, and 94 GHz. The comparisons of their corresponding reflectivities are shown in Fig. 1. As shown, the reflectivity values with 3 GHz are very similar to those with 13.6 GHz, indicating the Rayleigh scattering is satisfied for both frequencies in this application. Note the particle size simulated by global models at 1° scale is known much smaller than the reality, with diameters far smaller than 2.2 cm which leads to the similar reflectivity simulation for any frequency lower than 13.6 GHz.

Apparently, reflectivities at a frequency much higher than 13.6 GHz (such as 94 GHz, red cycles) would be a concern as mentioned by the reviewer.

For the attenuation, it would not cause a significant concern here as well, because 1) the model does simulate large particles that are enough to cause Mie scattering, and 2) the viewing geometry in COSP (from space to the ground) greatly relieve the attenuation from precipitation.

We agree with the reviewer that in real observation the conversion from Ku to S band is necessary. The reference listed is cited. In this application, it does not affect our results.

Please referrer to our response to the major comment 3 on the concern of "cloud detection".

We have rewritten the paragraph explaining the choice of 13.6 GHz, i.e., "The GPM radar frequency is higher than the NEXRAD (13.6 GHz vs. 3 GHz). Previous studies have shown conversions from Ku (13.6 GHz) to S band (3 GHz) are necessary when using GPM Ku band radar to calibrate the ground-based radars (Warren et al., 2018). Based on our previous study that quantitatively evaluated the coincident observations from NEXRAD and GPM over the CONUS, we found the 3D radar reflectivity fields obtained from the two independent platforms are highly consistent with each other after proper smoothing of GPM data in the vertical (Wang et al., 2019b). We performed a series of offline tests of COSP simulation using the frequency of 3 GHz (NEXRAD), 13.6 GHz (GPM Ku band), and 94 GHz (the cloud profiling radar onboard of the CloudSat satellite). Their corresponding reflectivities are compared in Fig. 1. As shown, the reflectivity values with 3 GHz are very similar to those with 13.6 GHz, indicating the Rayleigh scattering is satisfied for both frequencies in this application. Note the particle size simulated by global models at 1° scale is known much smaller than the reality (Marchand et al., 2009) with diameters far less than 2.2 cm (the wavelength of 13.6 GHz), which leads to the similar reflectivity simulation for any frequency lower than 13.6 GHz. To examine if the COSP can correctly handle the Mie scattering calculation, the frequency of 94 GHz used by the CloudSat is also tested, whose products have been widely used for the evaluation of coarse-resolution models (Zhang et al., 2010). As shown in Fig. 1, the reflectivities simulated with 94 GHz significantly deviate from those simulated with 3 GHz and 13.6 GHz when reflectivities > 10 dBZ, which reveals that the COSP simulator is capable of handling both Rayleigh and Mie scattering calculations. However, there is no difference using Ku band or S band in the COSP simulator in this study, because the simulated particles are too small to cause Mie scattering at these radar frequencies. An attenuation correction has been applied in case of existence of any large particles although they are extremely unlikely to occur in this application. Since the COSP mimics the satellite view from space to the ground, the layer below 1-km altitude is most vulnerable to the possible attenuation caused by large precipitation particles, which has been excluded from the comparison."

(b) It is not obvious that the implementation of 13.6 GHz in COSP is straightforward. The authors state that the simulator automatically uses Rayleigh scattering, but that cannot be appropriate under all circumstances, particularly if the focus is on convection. Attenuation will not only be significant below 1-km altitude: convective towers can cause attenuation in the ice phase as well. Similarly, the large hydrometeors found aloft may lead to Mie scattering. That the Rayleigh scattering assumption is inappropriate for the GPM PR has long been established in the literature, e.g.:
L'Ecuyer, T. S., and G. L. Stephens, 2002: An Estimation-Based Precipitation Retrieval Algorithm for Attenuating Radars. J. Appl. Meteor., 41, 272–285, https://doi.org/10.1175/1520-0450(2002)041<0272:AEBPRA>2.0.CO;2.
**We agree that in reality, convective towers can easily bring large ice particles aloft that lead to Mie scattering. However, in the coarse-resolution climate model, no such large particles are simulated.**

Having said that, it is entirely possible that the 13.6 GHz is a red herring here. If Rayleigh scattering is assumed, and no Mie scattering is included for large particles, the COSP calculation might as well be considered as if it were a 3 GHz radar. In that case, it is worth checking the COSP calculations for whether the frequency/wavelength matters.
**The 3 GHz and 13.6 GHz simulations have been checked, which are consistent.**

The authors have at least two options here. Either the authors provide corrected calculations following (for example) the papers above, for instance by applying such corrections to the COSP-simulated reflectivity. Alternatively, the authors develop a standalone forward simulator. If the latter, it is reasonable to assume Rayleigh reflectivity at 3 GHz (S-band) for comparison against the NEXRAD observations. Given the model microphysics assumptions (as listed in Table 1) it is relatively straightforward to calculate the Rayleigh reflectivity from the model ice and liquid water contents. This would be the most appropriate way to compare the model to the NEXRAD observations, but obviously requires some additional data processing, which may be difficult if the original cloud 3D fields were not included in the output.
**Since the two frequencies are proven to give the almost identical results in the model, we do not need to go with either of the options here.**

Major comment 2:

The lack of sufficiently high radar reflectivity aloft is concerning and while this could be a model bias, it would be helpful for the reader to have more information regarding the COSP calculations. In particular, in Table 1 the authors specify the density of ice and the distribution width. Following Morrison and Gettelman (2008), the remaining size distribution parameters lambda and N0 should be calculated from the mixing ratios directly. The COSP calculation will require the (consant) density of ice and distribution width as well as the (variable) lambda and N0, unless COSP has the appropriate information to calculate lambda and N0 itself from the mixing ratio. If COSP is not provided with the correct information, a constant lambda and N0 may be assumed by COSP, leading to erroneous calculations.
**Thanks for the good point. The lambda for hydrometeor size distribution in COSP is derived only from the mass mixing ratio, but the intercept parameter N0 is fixed in COSP. This is not consistent with MG2 yet. We have clarified this in the section describing COSP and also added discussion about the uncertainties from the simulator to the end of Section 3.1, that is, "although the simulated**

subgrid reflectivity distribution is improved by setting the microphysics assumptions used in COSP consistent with the MG2, the model is still significantly biased. In addition to the intrinsic model-observation differences in the number concentrations and mixing ratios of hydrometeors, there are other possible error sources related to the reflectivity calculation as mentioned in Section 2.2. For example, (1) the mixing ratios are not directly passed from the host model to COSP, instead are converted from the model's precipitation fluxes, (2) the spectral parameters for defining a Gamma distribution are not consistent from MG2, and (3) the assumptions of subgrid distribution and hydrometeor vertical overlap are simple and not consistent with other parts of the host model. In addition, the subgrid distribution results from COSP are calculated based on the assumption about the distribution of cloud and precipitation among the 50 subcolumns, which is independent of what E3SM uses. Therefore, a higher-order consistency between the COSP and the host model is warranted in future studies."

On a related note, if the E3SM has been evaluated against CloudSat and/or CALIPSO, that would provide helpful context to include about its ability to produce high-level cloud. While not compared with CloudSat/CALIPSO, the E3SM's high cloud fraction has been thoroughly evaluated using MODIS data product, where the model with the same configuration as this study agree well with the observation. Clarification has been added. "Differently from the model evaluation of cloud top height and high cloud fraction, where EAMv1 has shown good agreements with satellite observations over the CONUS, evaluation of radar echo top height indicates whether the processes internal to the cloud are producing precipitation correctly."

Major comment 3:

It is not clearly justified why the authors averaged their data to the 1-degree grid scale, when most of the information on the sub-grid scale is available to them. Averaging to 1-degree comes with its own problems (e.g. how to treat "cloud-free" regions) that may end up masking model deficiencies and it may have led to the disappearance of the characteristic CFAD shape in the NEXRAD analysis. Perhaps in Section 3.1, once the authors have performed their analysis using the sub-grid information, the authors could include some justification as to why the following analysis is done on the 1-degree averaged data. We agree that averaging to 1-deg comes with its own problems. However, 1 degree is the model's native grid spacing and it is important to evaluate how model performs at its resolution. In addition, the subcolumns are not what the model used in the simulations. It is just something that the simulator COSP independently assumes, which does not reflect how model performs at the subgrid scale. We have added the justification at the end of section 3.1, "in addition, the subgrid distribution results from COSP are calculated based on the assumption about the distribution of cloud and precipitation among the 50 subcolumns, which is independent of what E3SM uses. Therefore, a higher-order consistency between the COSP and the host model is warranted in future studies. In this following analysis, we focus on the evaluation of the simulated 3D radar reflectivity field at the model's native grid, which is 1-degree, since the subgrid information from COSP does not directly reflect how E3SM does it. Also, the convective cloud fraction in the ZM cumulus parameterization used in E3SM is fitted to be a function of cloud mass flux and should be viewed as a tunable

**parameter, whose evaluation is not very meaningful unless it becomes an independent variable, for instance, for grey-zone resolutions.”**

Minor comments:

1. Line 52-53: It is important to acknowledge in the introduction that these “convective processes” are not resolved by the model and that some sub-grid representation is needed. The evaluation can then be performed either on coarsened observations (as this study does) or on the sub-grid sampled model, as COSP does. Please include such a clarification in the introduction. **The information that convective processes are parameterized is already described in the E3SM model description section. The model does not include a sub-column sampler for subgrid clouds so it is difficult to evaluate the subgrid clouds of the model. COSP is just a diagnostic package and the subgrid information assumed in the COSP does not reflect how E3SM does it.**

2. Line 53-55: It is not obvious that these scattering effects are such an issue at S-band. Iguchi et al. (2018) consider the GPM-DPR which are smaller wavelength. At S-band, Rayleigh scattering could potentially be assumed which would make forward simulation much easier (easier than considering the sub-grid sampling for convection). Please rephrase this statement and consider studies using S-band radars specifically. [NB It should be noted that these lines are likely a result of the authors’ original intent to evaluate the model against GPM PR observations.] **See our response to the major comment 1.**

3. Line 69-72 and Line 98-99: It should be made clear to the reader that there is no difference in microphysics parameters between convective and stratiform (referring to Table 1). Some further clarification is then required regarding the sub-grid partition. Presumably, the model diagnoses a convective cloud fraction and a stratiform cloud fraction, with their respective water contents. These water contents may differ and therefore could lead to different simulated radar reflectivity. This is important information to include, so perhaps in Line 69-72 explain the convective-stratiform partition and in Line 98-99 clarify the typical differences between convective and stratiform water contents (noting that the microphysics parameters are the same). **The clarification of stratiform-convective partition in EAMv1 is added, “regarding the stratiform-convection partition, the MG2 stratiform cloud microphysics and CLUBB higher-order turbulence parameterization explicitly provide values for condensate mass and number, as well as an estimate of stratiform cloud fraction, whereas the convective cloud fraction is not parameterized in the ZM scheme, and is diagnosed from cloud mass flux for cloud radiation calculation, which is treated as a tunable parameter.”**
**However, those mixing ratios of different hydrometeor types at sub-grid scale are not directly input into the COSP. Instead, the COSP converts model-simulated precipitation fluxes to mixing ratios at sub-grid scale, which is inconsistent with the host model and could lead to different radar reflectivity simulated. The clarification has been added, “note the COSP does not use the hydrometeor mixing ratios from the host model to construct the particle size distribution (PSD) and then to calculate the radar reflectivity. Instead, it converts the model-simulated precipitation fluxes into mixing ratios before calling the radar simulation.”**

4. Line 106-108: The adoption of model-specific parameters is not unique and is a widely used approach when implementing COSP (or when developing their own forward simulator). Perhaps rephrase: “Following general usage of COSP, we modified the microphysics assumptions…”.

The Swales et al. (2018) paper explicitly mentions the need to "maintain consistency between COSP1 and the host model."
**Modifications have been made as suggested.**

5. Section 3.1: As stated above, the "out of the box" configuration of COSP is not advised and general use should always assume the model parameters. As such, it is recommended to remove the left-hand panels in Figure 2, as well as the standalone Figure 3, and focus instead on the differences between the model and observations from the right-hand panels. More specifically, in Figure 2: (1) Why does the x-axis start at 14 dBZ, when an 8 dBZ minimum reflectivity is considered? (2) What are the units of density? Per 2 dB (i.e. 2 dB bins)? (3) Why show these PDFs normalised? It should be important to note the absence of "cloud" above 8 dBZ as well. The authors should include a separate Figure showing the fraction of occurrence of Z>8 dBZ with height to compare this between model and observations.
**We prefer to keep the left-hand panels in Figure 3, as well as Figure 4. Although Swales et al. (2018) explicitly mentioned the need to maintain the consistency between the COSP and the host model, the impact hasn't been quantified. Figures 3 and 4 give a clear demonstration of the consequence. In addition, for the E3SM model community, it is good to show the problem.**

**For Figure 3, the minimum reflectivity should be 8 dBZ. The 14 dBZ was for the comparison with the GPM, which has been corrected. The figure has been modified and the interval of reflectivity bins is 1 dBZ.**

**Please note this is the subgrid distribution within $1°$ model grid elements, we use this comparison to explore the how the simulated subgrid reflectivity distribution from COSP differs from the observation. Following the referee's suggestion, we have added a figure showing the fraction of occurrence of $Z \geq 8$ dBZ with height (Fig. 6).**

**The related discussion has been added, "in addition to the mean values, the histograms of observed and simulated radar reflectivities are compared in for different altitudes, where the interval of reflectivity bins is 2 dBZ (Fig. 6). By comparing the occurrence of $Z \geq 8$ dBZ between model and observations, the model apparently has narrower distribution than the observations, and the model-observation deviation in maximum values increases with height. At 8 km and below, the model generally overestimates the sample sizes of smaller reflectivity values but lacks extreme high reflectivity values. However, at 11-km altitude, the model greatly underestimates the sample sizes of the entire reflectivity spectrum compared to the observation, causing the severe underestimation in the mean value."**

6. Section 3.2 and Figure 4: How is the mean calculated? In Section 2, we learn that for the instantaneous observation/simulated output, the mean is calculated in linear Z units (so that cloud-free areas are 0) and then converted to dBZ, with an 8 dBZ threshold. But how are values below 8 dBZ considered when calculating these long-term means? Or are the means (and standard deviation and 95th percentile) in-cloud only? In either case, it is useful to understand the occurrence of Z>8dBZ, so please add this to Table 2 and as a separate set of maps to complement Figure 4. The occurrence could help explain the difference in the mean, as the model could compensate for missing higher values by having a higher "cloud" occurrence.
**We discarded all the instantaneous grid boxes with $Z < 8$ dBZ. So, yes, the means, standard deviations, and 95[th] percentile values are all in-cloud samples only.**

**The sample numbers have been added to Table 2 as suggested, and we have added an additional figure for the sample numbers (Figure 6) as mentioned above.**

7. Section 3.3 and Figure 5: Again, normalization occurs in-cloud, so information is lost on the frequency of occurrence of cloud more generally. Could the authors comment on the difference in characteristic shape of the CFAD between NEXRAD and the model? The model follows the well-known shape with a maximum occurrence at dBZ that increases from cloud top to freezing level, and then slowly decreases or stays constant below the freezing level. That shape can be reproduced with NEXRAD, but it seems to have disappeared in the authors' analysis – is that solely due to the averaging to 1 degree? Perhaps the choice of Z for cloud-free regions is important here?

**As shown in Figure 6, the frequency of occurrence of cloud with $Z \geq 8$ dBZ are compared between the model and the observations.**
**We have added comment on the difference in characteristic shape of CFAD. "Regarding the overall shape of CFADs, the model follows the well-known pattern where the reflectivity value range of high frequency zone (> 3.2%) increases from cloud top to the freezing level, and then slowly decreases or remains constant below the freezing level. The cores of maximum frequency (> 5%) are located in the centres of the high frequency zones. However, these characteristics are not presented in the observations, whose high frequency zones are greatly skewed to the lower reflectivity values. The characteristics of NEXRAD's CFADs could be due to averaging from fine resolution (4 km) to coarse resolution (1°), as well as averaging of convective and stratiform components because the two components produce significantly different reflectivity profiles and magnitudes."**

**The choice of the threshold does not affect the shape of CFADs. In previous round of discussion, we have enlarged the sample size by using lower threshold of 0 dBZ, where the presentation of CFADs is not affected as shown in Figure R1.**

[Figure]

**Figure R1. The sensitivity test of changing minimum reflectivity threshold from 8 dBZ (a, b) to 0 dBZ (c, d).**

8. Line 219: "Above 11km, the model completely fails to simulate any reflectivity". There is some nuance here, as the authors use the 8-dBZ threshold. So: "Above 11km, the model fails to generate average reflectivity above 8 dBZ." Assuming that the authors have access to the data, it would be useful to report the typical reflectivity values that are generated by the model at these altitudes, even if below 8 dBZ.

**Modifications have been made as suggested. The reflectivity values simulated at 12 km is shown below. All the model data and observational data less than 0 dBZ were truncated during data processing. The typical reflectivity value would be 0-2 dBZ. We have added this information to the manuscript, "above 11 km, the model fails to generate average reflectivity above 8 dBZ, and the typical reflectivity value is between 0 and 2 dBZ at 12 km".**

[Figure]

**Figure R2. The histogram of simulated radar reflectivity at 12-km altitude.**

9. Line 238-240 and Figure 6: It is unclear what is actually being considered here. Is column-maximum reflectivity the maximum in a column on the 1-degree grid? What is then the "radar reflectivity" at a given "local time" in Figure 6? Is this an average over the entire CONUS, but only for grid boxes with this value above 8 dBZ? Or is it the maximum over the entire CONUS? The way this is calculated might partly explain the signals that appear, so all this needs to be clarified in the text.

**This figure has been deleted.**

10. Section 3.4 and Figure 7: As above, a general understanding of frequency of occurrence of Z>8dBZ would be useful in addition to these (normalised) diagrams.

**The sample sizes have been added to Table 2.**

11. Line 301-305: This conclusion should be removed, as should be the related Figures and discussion, as it is widely established that the microphysics assumptions of the forward simulator should be consistent with those in the host model (e.g. Swales et al., 2018).

**The figures and associated discussion help quantify the impact of inconsistent microphysics between the COSP and the host model. We prefer to keep these results.**

[revised manuscript text omitted]

Cumulus parameterization (rows: NBL restriction through cldfrc_dp1)

Microphysics parameterization (rows: prc_coef1 through thres_ice_snow)

**Figure List**

[Figure]

**Figure 1: Scatter plots between radar reflectivity values simulated by the COSP simulator at 3 GHz (x-axis) versus those simulated at 13.6 GHz (left y-axis) and 94 GHz (right y-axis).**

660

665

[Figure]

670

Figure 2: Examples of (a) original GridRad observation, (b) GridRad mapped over the E3SM model grid, and (c) the concurrent model simulation on 2016 May 11, 07:00 UTC, at the 2-km altitude.

675

**The Comparison of Radar Reflectivity Subgrid Distribution**

[Figure]

Figure 3: Comparison of radar reflectivity subgrid distribution between NEXRAD observations (red bars) and the simulations (blue bars) at the vertical levels of 2 km, 4 km, 8 km, and 11 km. Simulation results in the left and right columns are from the default microphysics assumptions in COSP and modified COSP microphysics assumptions, respectively.

[Figure]

**Figure 4: Scatter density plot between radar reflectivity values from the simulation with the modified microphysics assumptions (y-axis) versus those with the default microphysics assumptions (x-axis). The data shown are for April 2014. The dots are color labelled with their frequency of occurrence.**

685

690

[Figure]

**Figure 5: Plan view of radar reflectivity averaged from NEXRAD observations (a, d, g, j), EAMv1 simulation with the modified microphysics assumptions in COSP (b, e, h, k), as well as their absolute differences (c, f, i, l) at the level of 2-km, 4-km, 8-km, and**

700    **11-km altitude. The NEXRAD data are spatially averaged from native resolution to the model grid over 2014-2016 April-September period, and the simulation are vertically interpolated to the NEXRAD levels.**

[Figure]

705    **Figure 6: Comparison of radar reflectivity histograms at 1° scale between NEXRAD observations (red bars) and the simulations (blue bars) at the vertical levels of 2 km, 4 km, 8 km, and 11 km.**

[Figure]

**Figure 7: Contoured-Frequency-by-Altitude-Diagrams (CFADs) normalized by the total number of samples at all altitude levels for NEXRAD (a, d, g, j, m, p) and EAMv1 simulation with the modified microphysics assumptions in COSP (b, e, h, k, n, q) for the months from April to September averaged over 2014-2016 period. The box-whisker plots (c, f, i, l, o, r) for NEXRAD (red) and EAMv1(blue) are calculated using normalization at each individual level, where the center of the box represents the 50th percentile value, and the 25th and 75th percentiles are represented by the left and right boundary of the box, respectively. Whiskers correspond to the 5% and 95% values.**

[Figure]

 **Figure 8: Comparison of Contoured-Frequency-by-Altitude-Diagrams (CFADs) for the warm seasons over 2014-2016 between (a) NEXRAD, (b) EAMv1 simulation, and (c) the EAMv1-test simulation with reduced convective entrainment rate.**

---

## Author Response (AR3)

**Referee #4: Anonymous**

Review of "Using Radar Observations to Evaluate 3D Radar Echo Structure Simulated by the Global Model E3SM Version 1" by Jingyu Wang, Jiwen Fan, Robert A. Houze Jr, Stella R. Brodzik, Kai Zhang, Guang J. Zhang, and Po-Lun Ma.

The authors have provided a detailed and clear response to the reviews and the additions to the text are mostly good. However, the newly revealed details of treatment of the ZM scheme and its condensate in COSP has raised a major concern that must be addressed. Beyond that and some minor corrections, the manuscript may be considered for publication.

We thank the referee for the further detailed suggestions.

Major comment:

In the response to reviewers, and now also in L76-78, the authors provide the following information on treatment of convective cloud in their study: "…whereas the convective cloud fraction is not parameterized in mass flux-based ZM scheme (assumed to be <<1 for typical GCM resolutions such as at 1-degree grid spacing or coarser), and is diagnosed from cloud mass flux for cloud radiation calculation, which is treated as a tunable parameter."

Typical GCM evaluation studies considering precipitation combine both stratiform (or large-scale) precipitation as well as the output from the convection scheme (convective precipitation). Similarly, GCM evaluation studies considering clouds in terms of their radiative properties (e.g. outgoing longwave radiation) consider the grid-box mean OLR, that will be composed of both the convective and stratiform (or large-scale) radiative cloud. An evaluation study that does not take into account contributions from the convection scheme would vastly underestimate total precipitation and overestimate OLR from the GCM. The authors cite for instance the Xie et al. (2018) study which indeed considers convective contributions to precipitation and notes their importance.

The reviewer misunderstood the meaning of the sentence. We are evaluating the standard E3SM v1 and certainly the modeled precipitation and cloud radiative properties include the contributions from both convective and stratiform clouds, just like in Xie et al. (2018). Assuming convective cloud fraction to be <<1 in coarse resolution in GCMs is a typical and valid assumption and **it does not mean parameterized convective cloud is not included** in total precipitation and cloud radiative properties (as clarified, the convective fraction used in radiation is diagnosed from cloud mass flux). The added text was to clarify that it is not meaningful for evaluating convective cloud fraction (which was a major comment from the reviewer). In addition, our evaluation in this study is based on the COSP simulator which calculates the reflectivity for the combined cloud properties using its own subgrid assumption, so separating convective and stratiform cloud contributions to reflectivity is not possible. This is now further clarified in section 2.3 "Thus, COSP calculates the reflectivity for the combined cloud properties using its own subgrid assumption, and it does not distinguish convective and stratiform cloud contributions to reflectivity".

Similarly, the assumption that convective fraction is << 1 does not mean its impact on the results in this paper should be ignored. For 3D radar reflectivity, consider a grid-box with 5-10%

convective cloud. This cloud may reach high altitudes with significant amount of condensate, equivalent to Z=20 dBZ. That would lead to a grid-box mean greater than 0 dBZ and would show up in the authors' results. This may very well be the case in the GCM representation of MCS, which is brought up as a key point in the conclusions in L355-356: "This pattern suggests that the model is not adequately representing the mesoscale convective systems that dominate warm season rainfall in that region." Of course, in the interpretation of NEXRAD, no such convective elements are removed from the analysis, so the exclusion of convective condensate from the COSP analysis leads to a serious inconsistency in the evaluation.

As clarified above, convective cloud contribution is certainly included.

In brief:
1) I assume that condensate from the ZM scheme is not included in the COSP calculations. If this assumption is incorrect, the text will need to be adjusted to explicitly state how ZM convective cloud condensate is treated.

2) If my assumption is correct, however, the authors need to make clear in the paper how in a future study they could incorporate the ZM condensate. Presumably, using the same diagnosed cloud fraction as done in the radiation scheme would be the way forward. They will also need to make clear in L76-78 the potential shortcomings in this study for excluding convective cloud condensate.

Note that numerous studies using the CloudSat simulator incorporate convective cloud somehow in their analysis, starting with one of the first such studies: Bodas-Salcedo, A., Webb, M. J., Brooks, M. E., Ringer, M. A., William, K. D., Milton, S. F., and Wilson, D. R. (2008), Evaluating cloud systems in the Met Office global forecast model using simulated CloudSat radar reflectivities, J. Geophys. Res., 113, D00A13, doi:10.1029/2007JD009620.

Note also that the ZM changes the authors have made will only affect their simulated reflectivities indirectly, i.e. by causing changes in the stratiform condensate. It may be that if COSP were allowed to simulate reflectivity from convective condensate, much bigger changes would be seen.

As responded above, the reviewer's assumption is incorrect. All the major concerns from the reviewer were based on this incorrect assumption. We do not think there is any wordings in our paper indicating that parameterized convective clouds are not considered.

Minor comments:

L22-24: It is important to note in the abstract that this finding is in contrast with previous studies on cloud top height.
This piece of information has been added as suggested.

L105-107: "Note the COSP … radar simulation." This is an incorrect representation of COSP and should be removed. Depending on the COSP configuration used, it does use the hydrometeor mixing ratios (or the precipitation fluxes).

We agree that the COSP does use the mixing ratio to calculate the radar reflectivities, but it does not separately calculate the reflectivities for large-scale and convective at subgrid level. To clarify this point, we have reword the sentence, i.e.. "Thus, COSP calculates the reflectivity for the combined cloud properties using its own subgrid assumption, and it does not distinguish convective and stratiform cloud contributions to reflectivity".

L138-140: "Note the particle size… 13.6 GHz." and also L145: "because the simulated particles are too small to cause Mie scattering at these radar frequencies." This is an incorrect interpretation of GCM representation of cloud particle sizes and also is not supported by the Marchand et al. (2009) reference (which refers to effective radius, which is not relevant to this discussion). Most GCMs do assume a particle size distribution and thus implicitly contain particles of various sizes in their large-scale cloud and precipitation schemes. Such large particle sizes would therefore be considered by COSP. However, I do agree with the authors (thanks to their detailed response) that at 13.6 GHz such particles of similar size to the wavelength of 2.2 cm are going to be rare, particularly in the stratiform part of the cloud. The non-Rayleigh scattering regime would not be noticed until Z>40 dBZ, which is outside the regime shown in Figure 1. Please remove L138-140, which is incorrect, and rephrase L145 to finish (for example): "…COSP simulator in this study, because the simulated condensates are not large enough to lead to non-Rayleigh scattering, which is typically observed at Z>40 dBZ for Ku-band (Matrosov, 1992)."

Matrosov, S.Y., 1992. Radar reflectivity in snowfall. IEEE transactions on geoscience and remote sensing, 30(3), pp.454-461.

(The Matrosov reference Figure 2 shows reductions of 2-5 dB at the heaviest ice particle concentrations. More appropriate and more recent references can be found by the authors if they prefer.)

L138-140 has been removed as suggested. L145 has been modified and the reference has been added as suggested.

L193-196: It may be appropriate to add a 4th example here, pending the response to the major comment on the ZM convective condensate. Perhaps this means moving L201-204 here.

We modified the 1st example to better deliver the message about no separation of cloud types in COSP simulator, i.e., "the mixing ratios of hydrometeor types from different types of clouds are not directly passed from the host model to COSP, instead they are lumped together and equally divided among all the precipitating subcolumns."

L198: "Therefore, a higher-order consistency between the COSP and the host model is warranted in future studies." This statement ought to be revisited in the conclusions and the abstract, as it is a major point of this study.

This statement has been added to the abstract and the conclusions.

[revised manuscript text omitted]